# The molecular reach of antibodies crucially underpins their viral neutralisation capacity

Anna Huhn[1], Daniel Nissley[2], Daniel B. Wilson[1,3,10], Mikhail A. Kutuzov [1], Robert Donat [4], Tiong Kit Tan [4], Ying Zhang[3,11], Michael I. Barton [1], Chang Liu[5,6], Wanwisa Dejnirattisai[5,7], Piyada Supasa[5], Juthathip Mongkolsapaya [5], Alain Townsend[4,6], William James [8], Gavin Screaton [5,6,9], P. Anton van der Merwe [1], Charlotte M. Deane [2] ✉, Samuel A. Isaacson [3,12] ✉ & Omer Dushek [1,12] ✉

Key functions of antibodies, such as viral neutralisation, depend on high-affinity binding. However, viral neutralisation poorly correlates with antigen affinity for reasons that have been unclear. Here, we use a new mechanistic model of bivalent binding to study >45 patient-isolated IgG1 antibodies interacting with SARS-CoV-2 RBD surfaces. The model provides the standard monovalent affinity/kinetics and new bivalent parameters, including the molecular reach: the maximum antigen separation enabling bivalent binding. We find large variations in these parameters across antibodies, including reach variations (22–46 nm) that exceed the physical antibody size (~15 nm). By using antigens of different physical sizes, we show that these large molecular reaches are the result of both the antibody and antigen sizes. Although viral neutralisation correlates poorly with affinity, a striking correlation is observed with molecular reach. Indeed, the molecular reach explains differences in neutralisation for antibodies binding with the same affinity to the same RBD-epitope. Thus, antibodies within an isotype class binding the same antigen can display differences in molecular reach, substantially modulating their binding and functional properties.

Antibodies are multivalent molecules that contribute to immune responses by binding their antigens on the surfaces of pathogens. IgG antibodies usually have two identical antigen-binding fragments (Fabs) fused to a constant fragment (Fc). These Fabs enable antibodies to achieve high-affinity bivalent binding by simultaneously engaging two antigens. This is important because the monovalent Fab/antigen interaction is often too weak to be effective and indeed, bivalent antibodies are much more effective than monovalent Fabs at neutralising pathogens[1–5]. Although bivalent binding is important, we presently lack an understanding of the factors that influence it.

[1]Sir William Dunn School of Pathology, University of Oxford, Oxford, UK. [2]Oxford Protein Informatics Group, Department of Statistics, University of Oxford, Oxford, UK. [3]Department of Mathematics and Statistics, Boston University, Boston, Massachusetts, USA. [4]MRC Translate Immune Discovery Unit, MRC Weatherall Institute of Molecular Medicine, University of Oxford, Oxford, UK. [5]Wellcome Centre for Human Genetics, Nuffield Department of Medicine, University of Oxford, Oxford, UK. [6]Chinese Academy of Medical Science Oxford Institute, University of Oxford, Oxford, UK. [7]Division of Emerging Infectious Disease, Research Department, Faculty of Medicine Siriraj Hospital, Mahidol University, Bangkoknoi, Bangkok, Thailand. [8]James & Lillian Martin Centre, Sir William Dunn School of Pathology, University of Oxford, Oxford, UK. [9]Oxford University Hospitals NHS Foundation Trust, Oxford, Oxford, UK. [10]Present address: Kirby Institute, University of New South Wales, Sydney, New South Wales, Australia. [11]Present address: Department of Mathematics and Department of Biology, Northeastern University, Boston, Massachusetts, USA. [12]These authors contributed equally: Samuel A. Isaacson, Omer Dushek. ✉e-mail: deane@stats.ox.ac.uk; isaacsas@bu.edu; omer.dushek@path.ox.ac.uk

The ability of antibodies to bind an antigen surface depends on several factors. First, the monovalent on/off-rates determine initial antibody/antigen complex formation. Next, the antibody/antigen complex can bind a second antigen when it is within molecular reach, which is the maximum antigen separation that supports bivalent binding. The rate of this second reaction is heavily influenced by the antigen density[6–8]. When an antibody unbinds from one antigen, it can rebind the same or different antigen provided it is within reach and provided the epitope is not bound by another antibody. Although methods to analyse multivalent antibody/antigen interactions are available when both are in solution[9], we lack methods to analyse binding to surface-anchored antigen.

Existing methods to study antibody/antigen interactions have focused on isolating individual factors that contribute to overall binding. To determine monovalent kinetics, soluble monovalent antigen is injected over immobilised antibodies in surface plasmon resonance (SPR). A simple ordinary differential equation (ODE)-based binding model is fit to the monovalent SPR binding traces to determine the binding kinetics ($k_{on}$, $k_{off}$) and dissociation constant ($K_D$)[10]. However, monovalent parameters provide an incomplete understanding of antibody function[1]. While more physiological experiments can readily be performed by injecting antibodies over randomly-coupled surface antigens ('Bivalent SPR'), we lack methods to analyse the complex binding that results. To reduce this complexity and isolate the effect of reach, soluble antibodies can be injected over precisely spaced model antigens[11–13]. By measuring an apparent $K_D$ for different spacing, it has been estimated that antibodies can only bind antigen when spaced within ~16 nm, which is consistent with atomic force microscopy and structural studies[14]. However, we presently lack measurements of reach using physiological antigens making it difficult to know whether reach can predict antibody function. By removing the dependency on antigen density, current methods that isolate individual factors are unable to predict overall binding and hence the functional impact of antibodies in vivo, such as their ability to bind viral surface antigens at defined densities.

Here, we develop a method to mechanistically analyse bivalent SPR binding traces generated by soluble antibodies binding surface antigen. The method fits bivalent SPR data yielding accurate estimates of the monovalent binding parameters ($k_{on}$, $k_{off}$, $K_D$) and two additional biophysical parameters: the bivalent on-rate ($k_{on,b}$) and the molecular reach. Using this method, we investigate how multiple factors contribute to overall bivalent binding of patient-isolated antibodies specific for the receptor-binding-domain (RBD) of SARS-CoV-2.

## Results

### A particle-based model accurately fits bivalent SPR data highlighting the impact of molecular reach on antibody binding

We first used SPR to study the monovalent interaction between the IgG1 FD-11A antibody and RBD[15] (Fig. 1a, left). This standard method proceeds by injection of different concentrations of monovalent RBD over a surface immobilised with FD-11A. A monovalent ODE-based model is simultaneously fit to the entire monovalent SPR dataset, providing estimates of the $k_{on}$, $k_{off}$, and $K_D$ (Fig. 1b-left, c). We next reversed the orientation to study the bivalent interaction by injecting FD-11A at different concentrations over an RBD surface (Fig. 1a, Bivalent SPR). Standard SPR fitting software includes an ODE-based bivalent model that can seemingly fit bivalent SPR data (Fig. 1b, middle). This model adds a second bivalent binding step with a bivalent on-rate ($k_{on,b}$) and the same $k_{off}$. While the fit was not unreasonable, this model provides inaccurate $k_{off}$ and $K_D$ values that significantly differ from those produced by monovalent SPR, which is likely the reason it is seldom used (Fig. 1c).

A key assumption of the ODE-based bivalent model is that the molecules involved are 'well-mixed'. While this assumption is often used for chemical reactions in solution and it is reasonable for the first

antibody/antigen-binding step, it becomes unreasonable for the second step: once the antibody is bound to a surface immobilised antigen, the monovalent antibody/antigen complex can only bind a second antigen within reach whereas the well-mixed condition of the model assumes it can bind any free antigen on the surface. Moreover, the number of free antigens within reach is expected to be low and to decrease over time as more antibody binds. As a result, deterministic ODE models also fail to capture the local stochasticity of bivalent binding.

To address these limitations, we developed a more realistic stochastic and spatially resolved particle-based model of bivalent antibodies interacting with a random distribution of antigens ('particles') using the Gillespie method[16,17] (Fig. 1a, right). In this model, once antibodies bind to a surface antigen with the usual monovalent kinetics, the antibody/antigen complex can only bind a second antigen if it is within reach (with rate $k_{on,b}$ per antigen within reach). If multiple antigens are within reach, when an antibody unbinds one antigen, it can rebind another enabling antibodies to migrate on the surface. We used the model to simulate bivalent SPR traces to surfaces with the same (random) distribution of antigen but with three different reach distances (Fig. 1d). With a very short reach, antibodies could not bind a second antigen, resulting in monovalent binding with fast dissociation whereas increasing the reach allowed a larger fraction of antibodies to engage in bivalent binding, leading to much slower dissociation. This highlights the crucial role of molecular reach in determining antibody binding stability.

We developed a workflow to rapidly fit the particle-based model directly to bivalent SPR data (Fig. S1). This produced an excellent fit (Fig. 1b, right) and, unlike the ODE-based bivalent model, yielded $k_{on}$, $k_{off}$ and $K_D$ values in agreement with those obtained by monovalent SPR (Fig. 1c). Importantly, the model fit provided estimates of the bivalent on-rate and reach (Fig. 1e). We found the same binding parameters when analysing bivalent SPR with different levels of RBD on the chip surface, which confirms that the particle-model is correctly capturing how the antigen density impacts bivalent antibody binding (Fig. S2).

To further validate the model, we performed monovalent and bivalent SPR on four additional RBD antibodies (Fig. S3). We confirmed that the particle model correctly estimated the values of $k_{on}$, $k_{off}$, and $K_D$ across the 100-fold variation in affinity within these antibodies (Fig. 1f). As before, the particle model also provided estimates of the bivalent on-rate and the reach (Fig. 1g).

Finally, we repeated the analysis with an antibody that recognises a different antigen, namely CD19. Once again we found agreement between the ODE-based model analysing standard monovalent SPR and the particle-based model analysing bivalent SPR (Fig. S4).

In conclusion, a particle-based model accurately fits bivalent SPR data and allows measurement of binding parameters crucial for understanding bivalent antibody binding.

### The molecular reach is determined by both the antibody and the antigen

The molecular reach distances that our analysis produced for the five RBD antibodies (~38 nm, Fig. 1g) and the CD19 antibody (53 nm, Fig. S4) were notably larger than previous studies reporting that IgG1 antibodies can only bind bivalently when antigens were within ~16 nm[11,12,14]. They were also larger than estimates of the distance between the antigen-binding sites based on structural studies of whole antibodies (~15 nm)[1]. Furthermore, molecular dynamic (MD) simulations of the FD-11A antibody produced reaches of 3.46 to 17.58 nm with a mean of 13.05 nm (Fig. S5).

One possible explanation for these discrepancies is antigen size. Previous studies of molecular reach had used small model antigens (e.g., 4-hydroxy-3-iodo-5-nitrophenylacetate (NIP, 320 Da)[11], digoxin (780 Da)[14], 6x His-Tag (1100 Da)[12]) while we used much larger protein

 

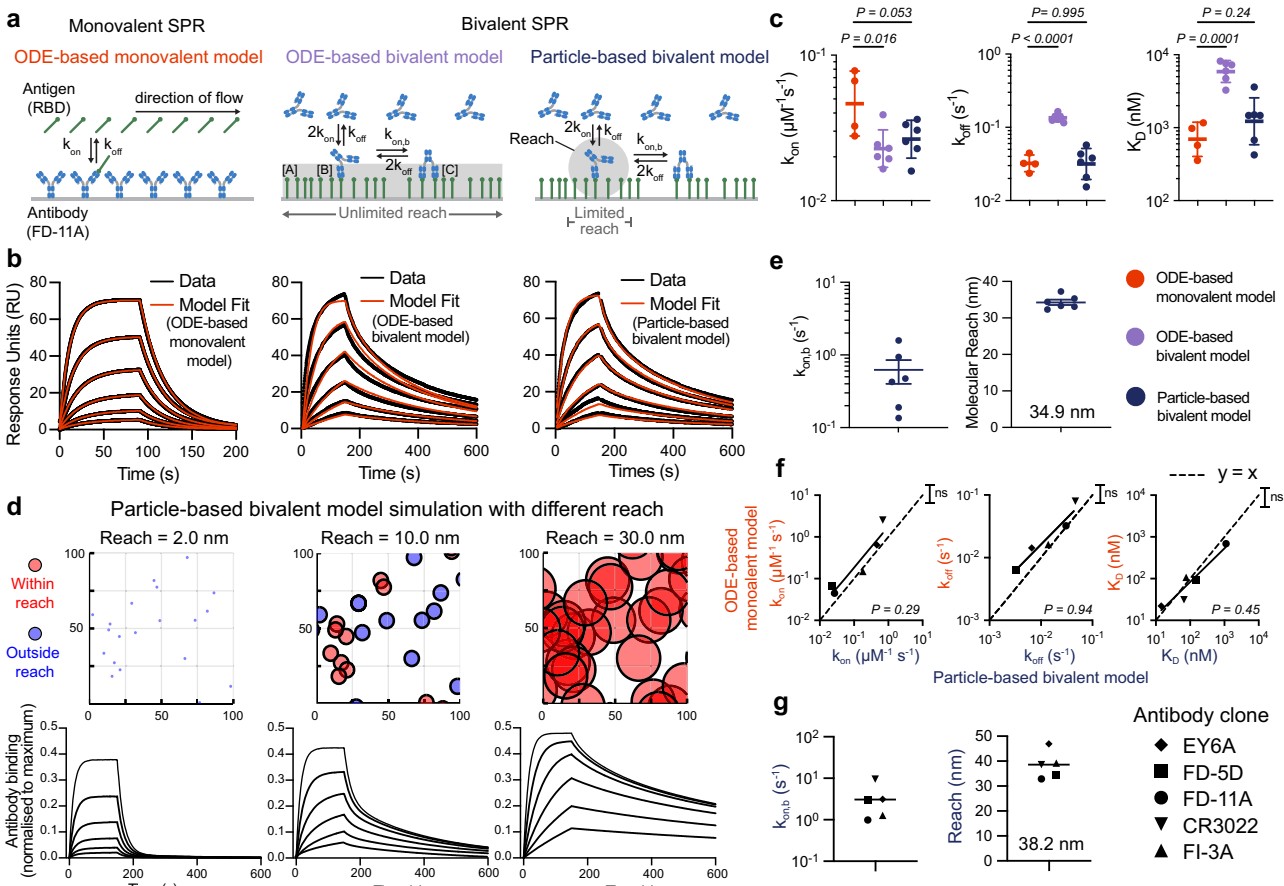

**Fig. 1 | A particle-based model for bivalent binding quantifies the impact of molecular reach and accurately analyses bivalent SPR traces. a** Schematic of the chemical reactions in models describing monovalent and bivalent SPR. The ODE-based and particle-based bivalent models assume that any free antigen on the surface or only free antigen within reach, respectively, can be bound by an antibody already bound to the surface with one arm (grey shaded regions). **b** Representative monovalent (left) or bivalent (middle, right) SPR traces using the antibody FD-11A interacting with RBD of SARs-CoV-2. The family of SPR traces are generated by 2-fold dilution of RBD starting at 2000 nM (left) or by a 2-fold dilution of FD-11A starting at 300 nM (middle, right−same data but different model fit). **c** The fitted binding parameters for monovalent ($N = 4$) and bivalent ($N = 6$) SPR experiments. Data are presented as geometric mean values ± SD. A two-way $t$-test with Dunnett's multiple comparison correction on log-transformed values was used to determine $p$-values. **d** Simulated SPR traces (bottom) for antibodies

injected over a surface with a random distribution of antigen (top) but with different values of reach. Antigens (circles) are coloured red if they are within reach of another antigen and blue otherwise. Parameter values: $k_{on} = 0.05$ μM⁻¹s⁻¹, $k_{off} = 0.05$ s⁻¹, $k_{on,b} = 1.0$ s⁻¹, and [RBD] = 0.0025 nm⁻². **e** The fitted bivalent binding parameters from the $N = 6$ bivalent SPR experiments. Data are presented as geometric mean values (for $k_{on,b}$) or mean values (for reach) ± SD. **f, g** Comparison of five antibodies analysed using monovalent and bivalent SPR from $N \geq 2$ independent experiments (EY6A: $N = 6$, FD-5D: $N = 6$, FD11A: $N = 8$, CR3022: $N = 5$, FI-3A: $N = 2$). **f** Comparison of the indicated parameter using both methods with the dashed line displaying perfect agreement ($y = x$). An F-test was used to determine a $p$-value for the null hypothesis that the dashed line and the fitted line to log-transformed binding parameters were equal. **g** The bivalent binding parameters. Source data for this figure are provided as a Source Data file.

---

antigens (RBD is 51,100 Da). To test this hypothesis, we injected an anti-phosphotyrosine antibody over a small phosphorylated peptide antigen coupled to polyethylene glycol (PEG) linkers comprised of 3 or 28 PEG repeats with a size of 2234 Da and 3336 Da, respectively (Figs. 2A, B, S6). Consistent with previous measurements using small antigens, our analysis produced a reach of 10.3 nm using the PEG3 linker and 13.4 nm using the PEG28 linker (Fig. 2B). Similar results were obtained at different PEG3 concentrations (Fig. S7). Using a polymer model, we calculated the theoretical increase in molecular reach between PEG3 and PEG28 to find agreement with the experimentally measured increase (Fig. 2C).

To determine if a ~38 nm reach was plausible for RBD binding antibodies, we used coarse-grained steered MD simulations. A coarse-grained representation of the FD-11A antibody bound to RBD was constructed, modified to include the Lys15 biotinylation site used to anchor RBD to the SPR chip surface (Fig. 3A). One Lys15 position was held fixed while the other was pulled away at constant velocity. We computed the fraction of native contacts at both paratope/epitope

interfaces (Fig. 3B) and the exerted pulling force (Fig. 3C). Unbinding events were identified based on a decrease in the fraction of native contacts, and this could be confirmed by visual inspection (Fig. 3D). At these events, the maximum distance between Lys15 on the two RBDs before the antibody unbound were recorded. This steered MD procedure allowed us to much more rapidly access extended conformations where FD-11A remained bound to both RBDs at large separation distances. In contrast, unrestrained simulations initialised at these separation distances would rarely access these extended conformations, which we expect are readily accessed in experiments that take place on the timescale of minutes.

We ran sets of simulations using different paratope/epitope interface strengths and as expected, increasing the strength allowed the simulation to explore bivalent binding conformations with a larger maximum RBD distance (Fig. 3E). Defining the molecular reach as the maximum distance achieved over different interface strengths, we obtained a value of 34.4 nm for FD-11A, which agrees well with the 34.9 nm reach estimated by bivalent SPR (Fig. 1e). We next repeated this

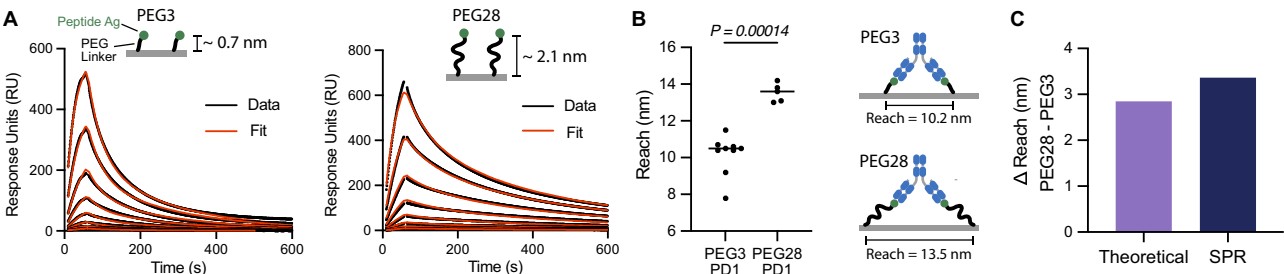

**Fig. 2 | The molecular reach is similar to the predicted antibody reach when the antigen is small. A** Representative SPR traces of the anti-phosphotyrosine antibody (PY20) injected over surfaces coupled to a small phosphorylated peptide linked to either 3 or 28 repeats of PEG. The antibody was injected at 8 concentrations (25 nM with 2-fold dilutions). **B** The fitted molecular reach ($N = 9$ for PEG3, $N = 5$ for PEG28 from independent experiments; see Fig S6 for other fitted parameters). An unpaired $t$-test on log-transformed values was used to determine $p$-values, corrected for multiple comparisons using the Holm-Šídák method. **C** Comparison of the difference in reach between PEG3 and PEG28 estimated by the worm-like-chain polymer model (see "Methods") and by experiments from panel (**B**). Source data are provided as a Source Data file.

procedure for the other antibodies we already characterised and an additional therapeutic antibody (REGN10987, Fig. S8) finding agreement with the SPR-determined reach (Fig. 3F).

Together, these findings demonstrate that our particle-based model provides accurate estimates of the molecular reach from bivalent binding data and that reach is the maximum separation distance between antigen anchoring points that support bivalent binding. Thus, the physical size of the antibody and the antigen contributes to the molecular reach.

We next examined the impact of antibody isotype on molecular reach. A previous study found that IgG1-4 sub-classes can tolerate a similar maximum antigen separation of ~16 nm[11], with a longer tolerance for IgM, which may result from its multimeric structure[11]. A study has suggested that IgA antibodies are more potent than IgG antibodies at neutralising SARS-CoV-2[18], raising the possibility that the IgA hinge may display different bivalent binding properties to IgG. To investigate this, we produced the FD-11A antibody as monomeric IgA and performed bivalent SPR on RBD surfaces. Interestingly, all binding parameters, including molecular reach, were similar between IgA and IgG1 FD-11A (Fig. S9). Taken together with previous work, this suggests that the antibody isotype does not appreciably impact the molecular reach or other antibody parameters in their monomeric form.

**The molecular reach of patient-isolated RBD-specific antibodies is the best correlate of SARS-CoV-2 neutralisation potency**

We next investigated the functional implications of molecular reach using a panel of 80 RBD-specific IgG1 mAbs previously isolated from SARS-CoV-2 infected individuals with known epitopes and viral neutralisation potencies[5] (Fig. 4A). The neutralisation potency is the antibody concentration required to produce 50% inhibition of infection ($IC_{50}$). We injected 3 concentrations of each antibody sequentially, analysing up to 32 antibodies in a single 48-h experiment (Fig. 4B). A high salt injection (3M MgCl) was used to regenerate the surface between antibody injections. In a pre-screen, 7 out of 80 antibodies remained bound after regeneration, and so were excluded from subsequent experiments. The FD-11A antibody was injected at the start and end of each of the 16 experiments that we performed to confirm that the antibody binding capacity of the surface remained largely intact over 48 h despite multiple regeneration steps (Fig. S10A, B).

We next fitted the particle-based bivalent model and the ODE-based monovalent model to the bivalent SPR data. We excluded 12 antibodies because they produced a poor fit to the particle-based model and 16 antibodies because the ODE-based model produced an accurate fit (Fig. S10C–G). We reasoned that an accurate fit by the ODE-based model meant that the bivalent SPR data contained no useful information about bivalent binding parameters. Possible explanations include an antibody that only binds monovalently because of

insufficient reach or an antibody that binds with very high affinity so that it does not unbind during the experiment. In these cases, it is not possible to quantify the increase in binding that bivalency provides.

The binding parameters for the remaining 45 antibodies displayed a 1000-fold variation in affinity, which was primarily the result of variations in the off-rate (Fig. 4C). Interestingly, the molecular reach exhibited large variations from 22 to 46 nm even though all antibodies shared the same IgG1 isotype and interacted with the same RBD antigen. The reach displayed some correlation with affinity and off-rate (Fig. 4D), which is consistent with previous work suggesting that higher-affinity antibodies can tolerate larger antigen distances when binding bivalently[11].

We found only modest correlations between neutralisation potency and the monovalent binding parameters (Fig. 5A–C). Given that binding parameters may independently contribute to predicting potency, we tried multiple linear regression, but the correlation was similar to the affinity alone (Fig. 5C vs D). Examining the additional bivalent parameters revealed no correlation with the bivalent on-rate (Fig. 5E), but the molecular reach displayed the best correlation of all single binding parameters (Fig. 5F). This increased further using a multiple linear regression model that included all the parameters (Fig. 5G). Together, this indicates that antibodies with a longer reach are better able to neutralise virus.

We reasoned that the ability of reach to predict neutralisation may be a result of its ability to predict bivalent binding and/or its ability to act as a proxy for the relevant blocking epitope. To test the latter, we computed the distance of each antibody epitope from a reference epitope in the ACE2 binding site (Fig. 5H). The approximate epitope location of these antibodies was previously determined using a competition assay and computational modelling[5]. As expected, we found that neutralisation potency was gradually reduced for antibodies that bound further from the blocking epitope interface (Fig. 5I). However, we found that the molecular reach did not correlate with this blocking epitope distance (Fig. 5J). We also used a previously reported epitope taxonomy but again, found no difference in molecular reach depending on epitope location (Fig. S11). This suggested that molecular reach predicts bivalent binding rather than the blocking epitope.

The observation that both epitope distance and reach correlated with neutralisation potency but not with each other suggested that each contained independent information relevant to viral neutralisation potency. We first confirmed that including the blocking epitope distance with the monovalent binding parameters improved the correlation (Fig. 5D vs K). In support of our hypothesis, we found that the correlation with all the binding parameters improved further when including the blocking epitope distance (Fig. 5G vs L).

Taken together, these results show that the best single-parameter predictor of neutralisation potency is the molecular reach and that

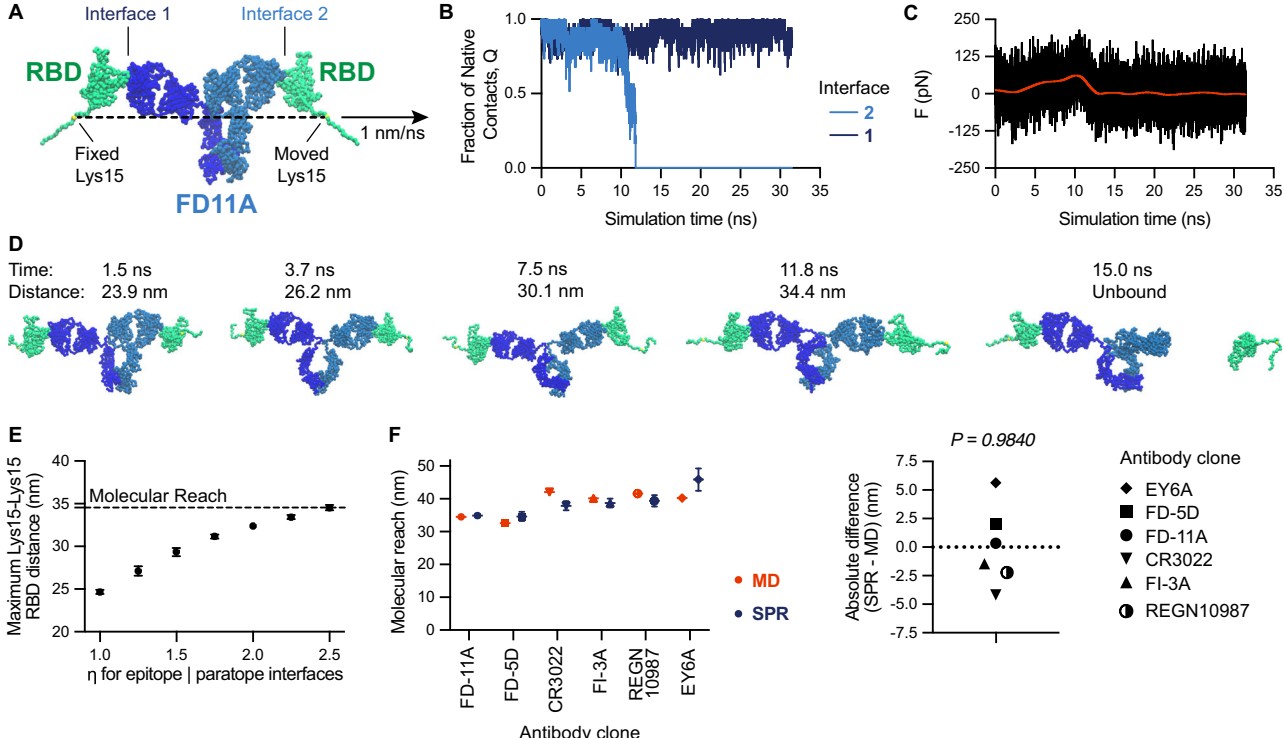

**Fig. 3 | Molecular dynamic simulations reproduce the molecular reach obtained by bivalent SPR highlighting the contribution of the antigen to reach. A** Coarse-grained structure of IgG1 FD-11A bound to two RBD antigens indicating the Lys15 biotinylation site (within an N-terminus AviTag), which anchors RBD to the SPR surface. The anchor point of RBD forming interface 1 was fixed while the one forming interface 2 was moved at a constant velocity. **B** The fraction of native contacts at the indicated interface (normalised to the number of contacts in the native structure) and **C** the force in the pulling direction exerted by the steering restraint (i.e., constant velocity over time). **D** Snapshots from the simulations at the indicated time points indicating the distance between Lys15 on each RBD. The maximum distance in this trajectory was 34.4 nm. **E** The maximum Lys15-Lys15 distance from $N = 50$ independent trajectories over the interface binding strength ($\eta$). The molecular reach is defined as the largest distance (horizontal dashed line). Mean values are shown, error bars are 95% confidence intervals computed from bootstrapping with 106 independent samples. **F** The molecular reach for the indicated antibody estimated by simulation or experiment (left) and the difference between these estimates (right). Reach values measured by SPR are shown as mean values ± SD determined from $N \geq 2$ independent experiments for each antibody (EY6A: $N = 6$, FD-5D: $N = 6$, FD11A: $N = 8$, CR3022: $N = 5$, FI-3A: $N = 2$, REGN10987: $N = 3$). A one-sample $t$-test is used to determine the $p$-value for the null hypothesis that the mean is 0 ($N = 6$). Source data are provided as a Source Data file.

increasing the molecular reach increases the neutralisation potency of RBD antibodies by enhancing bivalent binding.

## The binding potency of antibodies predicted by the particle model matches their neutralisation potency at antigen densities found on the virion

The neutralisation potency of an antibody emerges from multiple factors including its binding parameters ($k_{on}$, $k_{off}$, $k_{on,b}$, reach), the epitope, and the antigen density on the pathogen surface. We reasoned that the particle model could incorporate all of these factors to directly calculate an overall measure of antibody binding potency. To do this, we developed a workflow that used the particle model to simulate the amount of free antigen on a two-dimensional surface for different antibody concentrations from which the predicted binding potency (antibody concentration required to bind 50% of antigen) can be calculated (Fig. 6A).

We first used the FD-11A antibody to validate the workflow. We simulated the amount of free antigen at 60 min using the FD-11A binding parameters (Fig. 1c, e) for different FD-11A concentrations and for surfaces with 7 antigen densities (Fig. 6B). To compare these simulations with data, we mixed different concentrations of FD-11A IgG or Fab with live SARS-CoV-2 for 60 min before adding it to Vero cells and determined infection at 20 h (Fig. 6C). We compared the predicted binding potency with the experimental neutralisation potency (Fig. 6D).

At the lowest antigen densities, the model predicted a poor binding potency of ≈600 nM that matched the expected affinity of a bivalent antibody that can only bind a single antigen ($K_D/2 = 614$ nM, where the factor of 2 accounts for the two antibody Fabs and $K_D = 1228$ nM is the monovalent affinity). This is because the average distance between antigens at this density (312 ± 163 nm) is much larger than the molecular reach of FD-11A (34.9 nm) impairing bivalent binding. The predicted binding potency improved by >100-fold as the antigen density increased enabling FD-11A to bind bivalently and eventually reaching the experimental potency. The Fab failed to neutralise virus even at concentrations above 3000 nM, which likely reflects the fact that the monovalent antibody/RBD affinity is much lower than the ACE2/RBD affinity ($K_D \sim 75$ nM[19]).

We next used the workflow to predict binding potency at different antigen concentrations for all 45 antibodies. The predicted and experimental potencies were similar at intermediate antigen densities of 0.0005–0.001 nm$^{-2}$ (Fig. S12) and this could be further improved if only a subset of 24 antibodies were included that bound near the blocking epitope (Fig. 6E). At very low or very high antigen densities the predicted IC$_{50}$ was heavily over-estimated for potent antibodies or heavily under-estimated for low potency antibodies, respectively (Fig. 6E, left and centre panels). At these extreme densities, the molecular reach of each antibody is irrelevant for the model predictions. At the low densities, the antigens are spaced so far apart (312 ± 163 nm) that even antibodies with the longest reach fail to

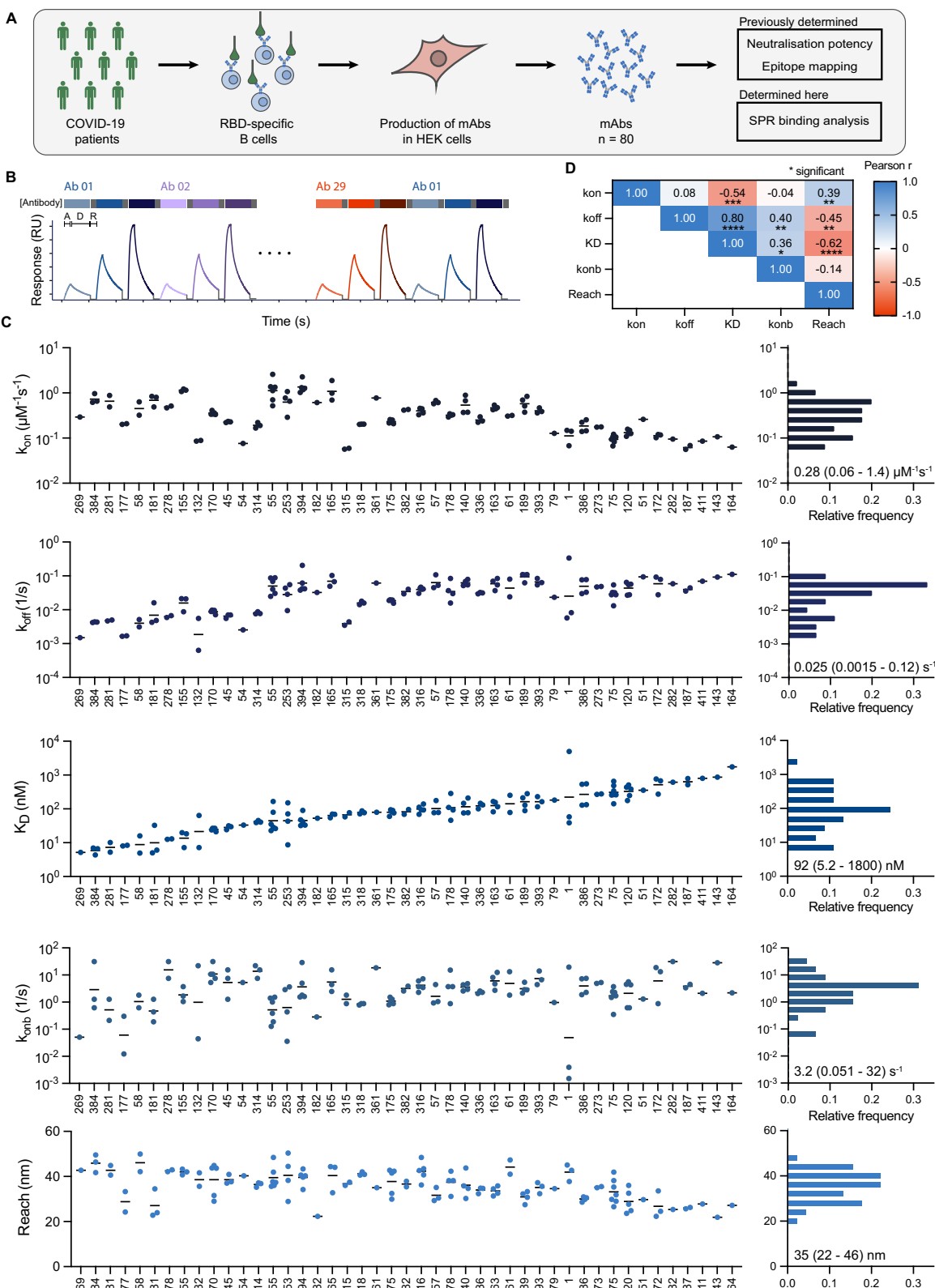

**Fig. 4 | High-throughput analysis of patient-isolated antibodies using bivalent SPR reveals large 22–46 nm variations in reach.** **A** Schematic of antibody identification and characterisation. **B** Workflow for high-throughput bivalent SPR indicating that each antibody was injected at 3 concentrations with association (A, 150 s), dissociation (D, 450 s), and surface regeneration (R) steps. **C** Complete set of binding parameters for 45 antibodies ordered by affinity (left) or displayed as distributions (right–mean with min/max shown, $N = 45$). **D** Parameter correlations. Pearson correlation coefficients were computed for every parameter pair using average parameters for each antibody. Values were log-transformed for $k_{on}$, $k_{off}$, $K_D$ and $k_{on,b}$ ($N = 45$), and two-tailed $p$-values were calculated. Source data are provided as a Source Data file.

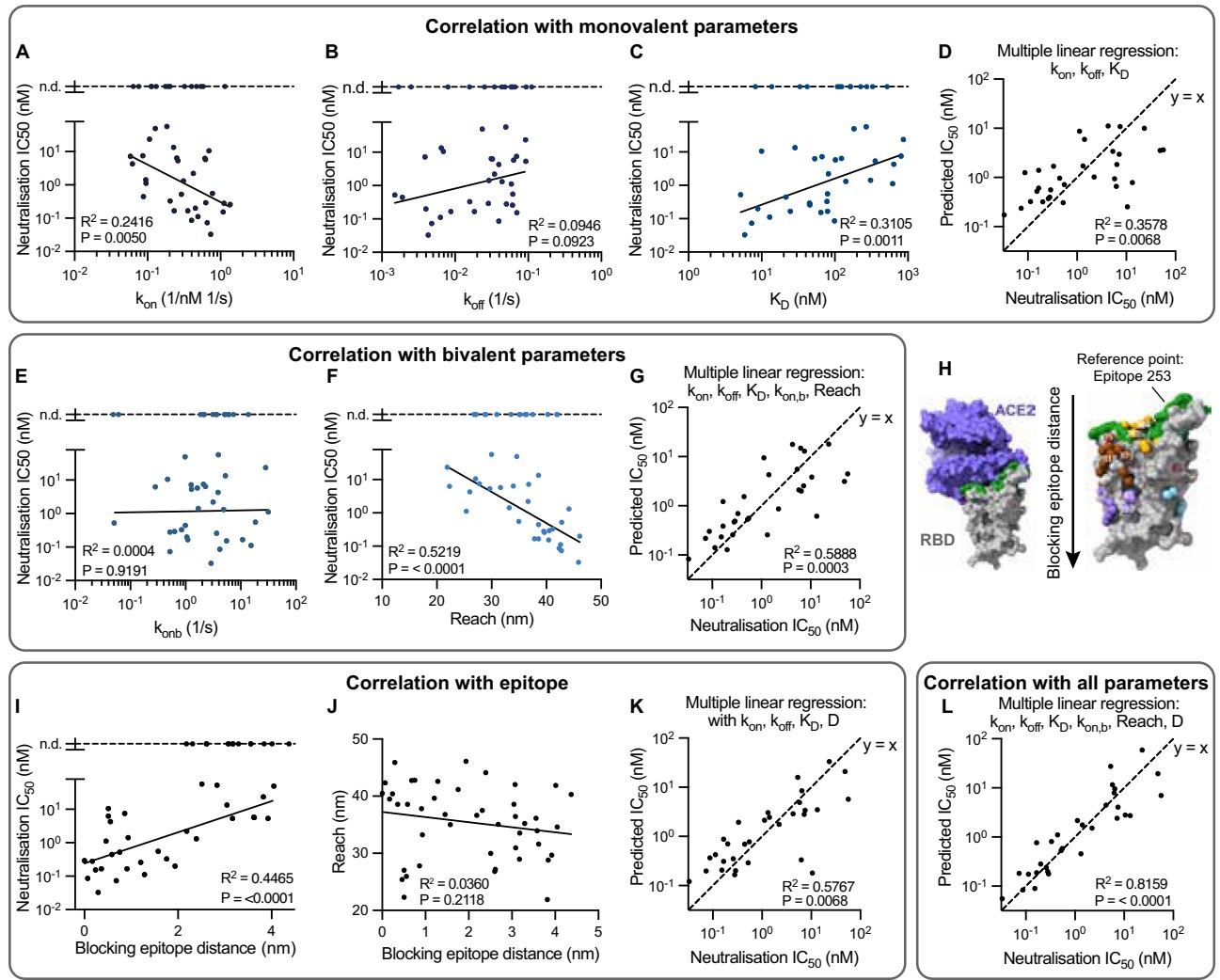

**Fig. 5 | The molecular reach of RBD-specific antibodies correlates with neutralisation potency but not with the ACE2 blocking epitope. A–G** Single and multiple linear correlations between the indicated parameters of 45 antibodies (individual data points) and their neutralisation potency ($IC_{50}$), the concentration of antibody required to reduce in vitro SARS-CoV-2 infection by 50%. The 13 antibodies whose potency was not determined (n.d) were excluded from the fit (top dashed line). **H** Structure of RBD showing the ACE2 binding site (left, blocking epitope in green) and the epitope of each antibody on the structure (right). The blocking epitope distance is calculated using the epitope for antibody 253, which is furthest at the top of RBD, as the reference point. Images are taken from ref. 5. **I, J** Correlation between blocking epitope distance and (**I**) neutralisation potency or (**J**) reach. **K, L** Multiple regression with the blocking epitope distance using (**K**) only the monovalent binding parameters or (**L**) all the binding parameters. The neutralisation potency and epitope locations were previously determined[5]. Multiple linear regression included only main effects and all correlations were performed on log-transformed parameters except for molecular reach and epitope distance. An F-test was used to determine a *p*-value for the null hypothesis that the slope of the fitted line is equal to zero. Source data are provided as a Source Data file.

achieve bivalent binding, under-estimating their predicted potency. Conversely, at very high densities, the antigens are spaced so close (2.5 ± 1.3 nm) that even antibodies with the shortest reach achieve bivalent binding, overestimating their predicted potency. At intermediate densities (~0.0005 nm⁻²) corresponding to a mean antigen spacing of 22 ± 11 nm, efficient bivalent binding is achieved only by the subset of antibodies with longer reaches, and it is at these intermediate densities that the predicted binding potency matches the experimental neutralisation potency (Fig. 6E, right panel). Importantly, these intermediate densities are similar to the estimated density of the Spike protein on the surface of SARS-CoV-2 (Fig. 6F).

Finally, we wondered if the predicted binding potency can explain discrepancies in neutralisation potency based on monovalent binding and epitope location. We identified two antibodies in our dataset that displayed large differences in neutralisation potency despite binding with similar monovalent affinities to the same RBD-epitope (Fig. 7A). Given that these antibodies also displayed differences in their bivalent binding parameters, we calculated their binding potency at different

antigen densities finding that the model correctly differentiated between them only at intermediate densities found on the virion (Fig. 7B, C). Therefore, antibodies with a similar affinity to the same epitope can display large differences in function because of differences in their molecular reach, and hence propensity for bivalent binding.

Together, these results underline the importance of molecular reach and the antigen density/spacing in determining bivalent binding and hence antibody function.

## Discussion
The challenge of studying antibody binding has motivated experimental methods to isolate individual factors that contribute to antibody binding. Experimental methods are routinely used to study monovalent antibody/antigen kinetics and methods are now available to measure reach for precisely spaced model antigens[11–13]. Although mathematical methods are available to study bivalent binding in certain limits, such as soluble antibody/antigen

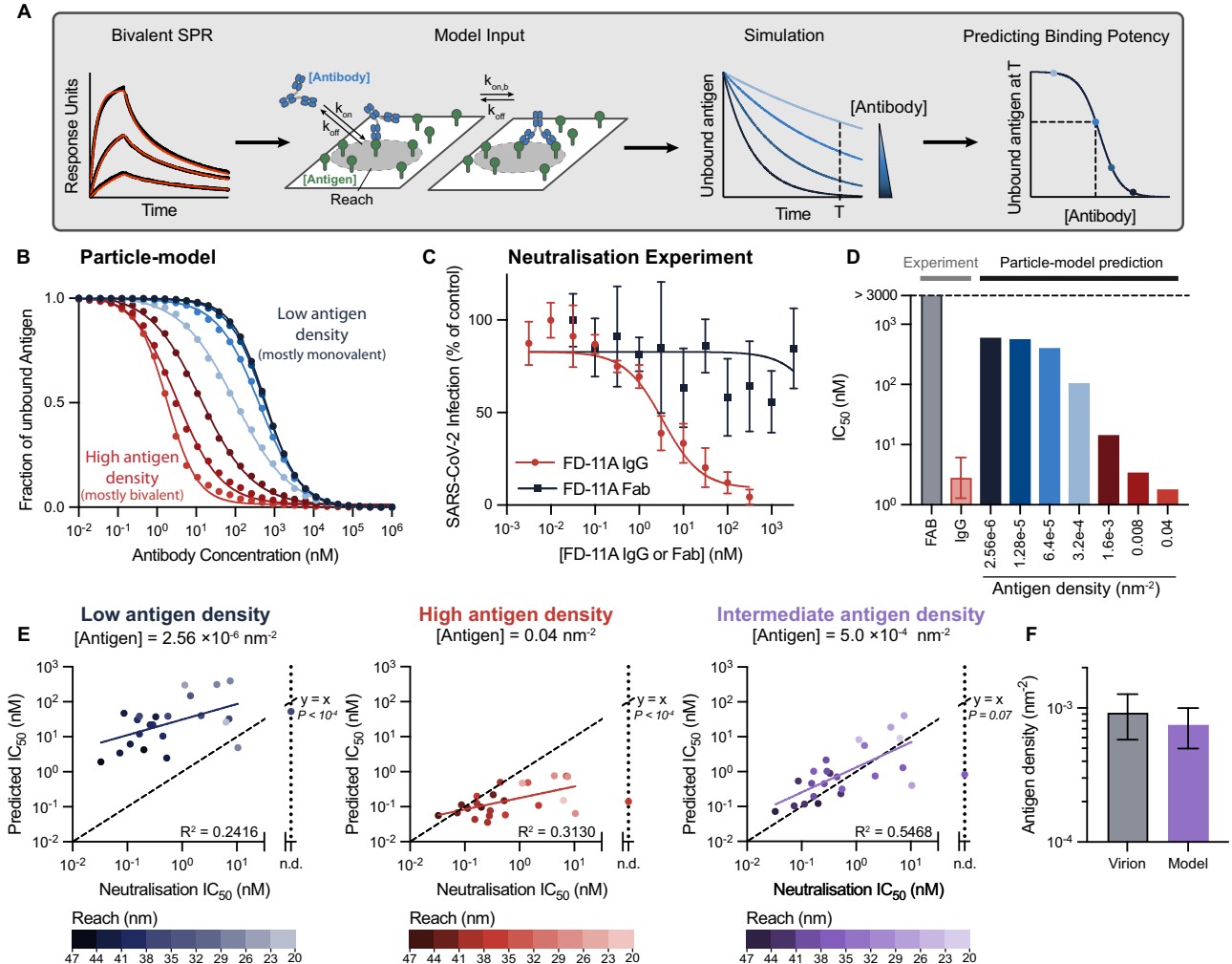

**Fig. 6 | The antibody binding potency calculated by the particle model equals the antibody neutralisation potency determined by live virus experiments.**
**A** Schematic of workflow to determine the predicted antibody binding potency defined as the concentration of antibody required to bind 50% of antigen randomly distributed on a two-dimensional surface. The model input includes the antibody/antigen concentrations and binding parameters. **B** Simulations of the fraction of free antigen after 60 min using the FD-11A antibody binding parameters. **C** The ability of live SARS-CoV-2 virions to infect target cells when pre-mixed for 60 min with different concentrations of FD-11A IgG (red) or Fab (black) determined at 20 h. Data from 4 biological replicates are presented as mean values ± SD. **D** Fitted potency values ($IC_{50}$) from **B** simulations and **C** experiments. Error bars indicate 95% confidence intervals of fitted $IC_{50}$ value. **E** Comparisons of the predicted binding potency over the experimental neutralisation potency for 25 antibodies that bind within 2.37 nm of the blocking epitope for the indicated antigen concentrations (see Fig. S12 for all antibodies and all antigen densities tested). A linear fit on log-transformed $IC_{50}$ values (solid line) is compared to the identity line ($y = x$) using an F-test. The shading of each point indicates the molecular reach of the respective antibody. **F** Comparison of the Spike density on SARS-CoV-2 virion[22] (Mean value ± SD) with the intermediate antigen density producing absolute agreement using the particle-model (0.00075 nm$^{-2}$ [0.0005, 0.001]). Source data are provided as a Source Data file.

interactions, a method for analysing antibody binding to a random distribution of anchored antigen is presently unavailable[7,9,20,21]. Here, we developed a fast spatial and stochastic particle-based method to directly fit antibody binding to random antigens at any density enabling estimates of monovalent kinetics/affinity, the bivalent on-rate, and the molecular reach.

We validated the particle-based model in several ways. First, the model correctly identifies the monovalent kinetics and affinities for several antibodies across a large variation in affinity using two antigens: RBD (Fig. 1) and CD19 (Fig. S4). Second, the model identifies the same parameter values at different antigen densities on the SPR surface (Figs. S2, S7), highlighting that it can accurately capture how antibody binding depends on antigen density. Lastly, MD simulations reproduce the fitted molecular reach values obtained by bivalent SPR (Fig. 3).

By analysing a large number of SARS-CoV-2 RBD-specific antibodies, we found that the molecular reach (up to 46 nm) is much larger

than previous reports (up to ~16 nm)[11,12,14]. We have shown this to be the result of differences in antigen sizes, with previous reports focused on low molecular weight model antigens (<1100 Da) compared to the RBD antigen we have used (51 kDa). This suggests that antibodies can simultaneously bind two antigens anchored much further apart than the physical size of an antibody provided that the antigen is large and can display tilting flexibility relative to the surface, which is the case for surface RBD/Spike[22] (Fig. 8). Moreover, the large variation in reach across antibodies with the same IgG1 isotype that we report (22–46 nm) implies that a universal reach value is unlikely and that this critical parameter will need to be assessed for each antibody/antigen combination. The importance of the molecular reach is underlined by the observation that it was the best single-parameter predictor of viral neutralisation, with a correlation larger than the monovalent binding parameters and the blocking epitope. This is consistent with molecular reach being a proxy for bivalent binding, which itself is known to be important for SARS-CoV-2 neutralisation[3,5].

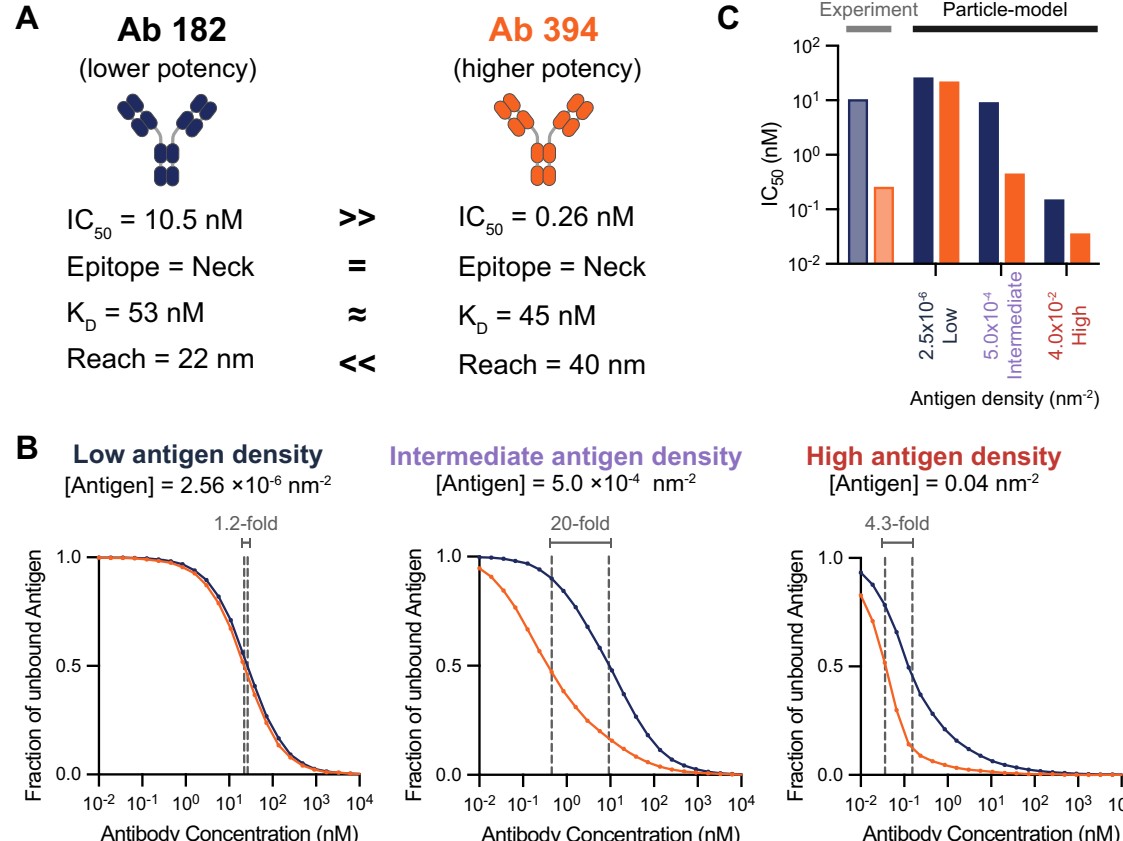

**Fig. 7 | Antibodies with similar monovalent affinity to a shared epitope can display large differences in neutralisation potency because of differences in propensity for bivalent binding. A** The experimental neutralisation and binding parameters of antibody 182 and 394. **B** Simulations of antibody 181 (blue) and 394 (red) binding after 60 min using their measured binding parameters (vertical dashed lines indicate $IC_{50}$ binding potency). **C** Comparison of predicted binding potency and experimental neutralisation potency. Source data are provided as a Source Data file.

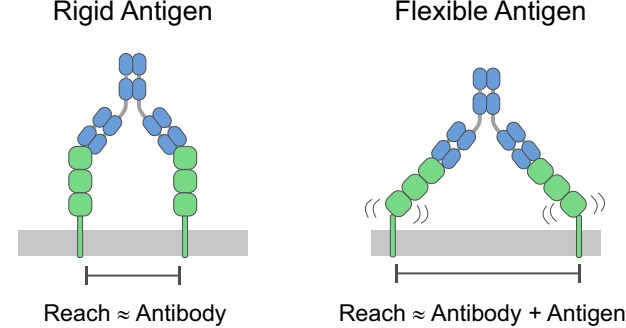

**Fig. 8 | The molecular reach is expected to exceed the antibody reach when the antigen contains flexible elements.** The molecular reach is expected to largely rely on the antibody reach when binding to rigid antigens (left). However, when the antigen is flexible (e.g. through a flexible hinge) then the overall reach depends on both the antibody and the antigen (right). In other words, antibodies can bind bivalently to flexible antigens anchored at distances that are larger than the reach of the antibody.

Previous work has shown that the maximum antigen separation that can be tolerated by monomeric IgM, IgE, and different IgG sub-classes binding to a model antigen is similar (~16 nm)[11,13] and we have shown here that IgG1 and IgA have similar reaches. While these results suggest that different antibody isotypes have the same maximum reach, they may differ in their minimum reach (minimum antigen separation distance), as previously suggested[11]. Antibody isotype does affect the binding strength of the overall antibody binding to

multivalent antigen, also termed antibody avidity[23]. The major mechanism for this is isotype-dependent differences in valency, with IgM antibodies having 10–12 antigen-binding sites, IgA having 2–4, and IgD, IgE and IgG typically having 2[23]. Interestingly, IgG4 has been shown to become monovalent and bi-specific after secretion as a result of antibody arm exchange through labile hinge disulfide bonds[24].

A unique feature of our analysis is that the complete set of binding parameters can be used to quantitatively predict antibody binding. This enables estimates of the antibody concentration required to bind 50% antigen on surfaces with different antigen densities. Interestingly, the predicted binding and experimental neutralisation potencies matched only when using the antigen density of Spike on the surface of SARS-CoV-2 in our simulations. At this density, the mean distance between antigens overlapped with the measured molecular reaches. A limitation of our method is that these calculations relied on monomeric antigen but the Spike protein is trimeric and contains three RBDs that are closer together than the molecular reaches that we report. This apparent discrepancy can be accounted for if the predominate mechanism of bivalent binding involves two RBDs on different Spike trimers. Although there is evidence that some antibodies can bind two RBDs within a Spike trimer[4,25], a Spike trimer typically contains only a single accessible RBD[22] and the lifetime of antibodies depends on the Spike concentration implying inter-Spike binding[18]. Moreover, directly resolving antibody/Spike binding revealed Spike cross-linking[26]. Therefore, our results are consistent with the present antibodies predominately binding RBD across Spike trimers.

While we have shown that the molecular reach of RBD antibodies is crucially important for the neutralisation of SARS-CoV-2, it remains to be shown whether this applies to other viruses. Molecular reach is

most likely to be important when antibody is limiting and/or monovalent binding affinity is low. In contrast, reach will likely be less important when monovalent affinity is high and/or when antibody is not limiting, such as antibody-dependent cellular cytotoxicity, antibody-dependent phagocytosis, or antibody-mediated complement activation[23]. In addition, the molecular reach may be less important when the antigen density and/or mobility is high. While antigen density is easily and frequently measured, it is more difficult to measure antigen mobility, especially on viral surfaces. Importantly, a longer reach may reduce bivalent binding if the free antibody arm explores a larger volume away from the antigen, analogous to the impact of reach on membrane-confined enzymatic reactions[27].

Our findings have implications for vaccine design and assessment of the induced antibodies. A key objective of vaccination is to induce neutralising antibodies[28]. When bivalent binding to the pathogen surface is required for neutralisation, optimally designed vaccines may need to reproduce antigen size, flexibility, and spacing on the pathogen surface[29–32]. Furthermore, assessment of the quality of infection- or vaccine-induced antibodies could include an analysis of bivalent binding, including assessment of the molecular reach, as this information can then be used to quantify the ability of antibodies to neutralise the virus and hence identify the level of protection achieved within the individual. Thus, we suggest that improvements in vaccine design and assessment can be achieved by analysis of bivalent binding of antibodies.

In conclusion, we provide a validated method to analyse bivalent SPR data and to predict antibody binding to antigen surfaces at any density. Although we have focused on using SPR, the method can be used to analyse any bivalent molecule using any instrument that measures binding. While existing methods have largely focused on studying and optimising the monovalent paratope/epitope interface, our method now provides the ability to easily analyse and predict bivalent binding. This should improve the ability to study and engineer antibodies and other native and synthetic bivalent molecules.

## Methods
### Proteins
**Production of monovalent streptavidin.** Monovalent Streptavidin was produced using a method previously described to generate streptavidin tetramers of defined valency fused to Spycatcher[33]. Two different streptavidin subunits were used: Streptavidin-SpyCatcher, which contains a functional streptavidin monomer fused to SpyCatcher at its C-terminus and a 'dead' streptavidin, which contains a mutation in the streptavidin monomer that has negligible biotin-binding activity. Individual subunits were expressed in *E. coli* BL21-CodonPlus (DE3)-RIPL cells and refolded from inclusion bodies. Inclusion bodies were washed in BugBuster (Merck Millipore 70921) supplemented with lysozyme, protease inhibitors, DNase I, and magnesium sulfate as per the manufacturers' instructions. To obtain monovalent streptavidin-SpyCatcher, the subunits were mixed at a 3:1 molar ratio of dead streptavidin to streptavidin-SpyCatcher. Tetramers were refolded by rapid dilution and precipitated using ammonium sulfate precipitation. Precipitated protein was resuspended in 20 mM Tris (pH 8.0), filtered (0.22-μm filter), and loaded onto a Mono Q HR 5/5 column (GE Healthcare Life Sciences). Desired tetramers were eluted using a linear gradient of 0−0.5 M NaCl in 20 mM Tris (pH 8.0), concentrated, and buffer exchanged into 20 mM MES, 140 mM NaCl (pH 6.0).

**Production of biotinylated-RBD.** Affinity-purified SARS-CoV-2 (Wuhan) RBD was biotinylated using EZ-Link Sulfo-NHS-LC-biotin (Life Technologies, USA, A39257) according to manufacturing protocol. Biotinylated-RBD was subjected to Zeba™ Spin Desalting Columns (7k MWCO) (Thermo Scientific, USA, catalogue 89889) to remove excess biotin.

**Production of FD-11A, FD-5D, EY-6A, FI-3A, CR3022, REGN10987 antibodies.** Monoclonal antibodies were produced as previously described[15,34]. Briefly, expression plasmids were transfected into the ExpiCHO cell lines according to the manufacturer's protocol (Thermo Fisher). Supernatant containing monoclonal antibodies were clarified by centrifugation (1400×*g*, 5 min) and 0.45 μM filtered before purification. Monoclonal antibodies were affinity purified using a MabSelect SuRe (Cytiva) pre-packed column. Purified mAbs were then desalted using Zeba Spin Desalting Column (ThermoFisher) or diafiltered using an Amicon Ultracentrifugation Column (50k MWCO).

To produce IgA from IgG1 FD-11A, the sequence encoding the VH region of the FD-11A antibody was fused to the human IgA1 Constant region on a pCDNA3.1 backbone. The light chain sequence of FD-11A IgA1 and IgG are identical. The plasmids encoding the FD-11A IgA heavy chain and light chain were co-transfected into Expi293 cells according to the manufacturer's instructions. FD-11A IgA1 was affinity purified using Peptide M agarose (InvivoGen) according to the manufacturer's instructions. The FD-11A IgA1 was diafiltered into PBS using a 100k MWCO Amicon filter and purified using size exclusion chromatography.

**Production of 80 mAbs specific for RBD.** Antibodies for high-throughput screen used in the present study were produced previously[5]. All antibodies contain the human IgG1 backbone paired with either $\kappa$ or $\lambda$ light chains.

**Production of Spytag-CD19 antigen and antibody.** Spytag-CD19 fused to Spytag was produced as previously described[35]. Briefly, the extracellular domain of CD19 is expressed as a fusion protein with Spytag and Histag fused to its C-terminus, and the Sumo protein via a HRV cleavage site fused to its N-terminus (Sumo-HRV-CD19-Spytag-Histag). The protein SUMO was used to stabilise Spytag-CD19 during production. A CMV expression plasmid encoding Sumo-HRV-CD19-Spytag-Histag was transfected into Expi293F Cells (ThermoFisher) using the ExpiFectamine 293 Transfection Kit (ThermoFisher Scientific, A14524). Cells were incubated for 4−5 days for protein expression. Following, the supernatant was harvested, and in the first step, the fusion protein was purified using Ni-NTA Agarose column. Next, the protein was concentrated and loaded onto a Superdex 200 10/300 GL (Cytiva, 17-5175-01) size exclusion chromatography column. Next, HRV 3C Protease Solution Kit (Pierce™, 88946) was used to cleave SUMO from the CD19 fusion protein. The protease was removed via Glutathione Agarose (Pierce™, 16100), followed by Ni-NTA Agarose to remove SUMO. Finally, the protein was stored in size exclusion buffer (25 mM NaH$_2$PO$_4$ and 150 mM NaCl at pH 7.5) and frozen in suitable aliquots at −80 °C.

The anti-CD19 antibody (clone SJ25C1) was purchased from BioLegend (cat no. 363001).

**Production of PEG-coupled phosphorylated peptide antigens and antibody.** Phosphorylated peptide antigens were a custom commercial order (Protein Peptide Research Ltd, UK) with the following sequences: Bio-(PEG)$_3$-SVPEQTEY*ATIVFPSG (PEG3) and Bio-(PEG)$_{28}$-SVPEQTEY*ATIVFPSG (PEG28), where Bio indicates biotin, PEG$_X$ is X repeats of polyethylene glycol, and * indicates phosphorylation.

The anti-phosphotyrosine antibody (clone PY20) was purchased from Absolute Antibody (Ab00294-1.1).

### Surface plasmon resonance
**Monovalent SPR for RBD antibodies.** A BIAcore 8K (Cytiva) was used to measure the affinity and the kinetics of soluble RBD to immobilised antibody. Approximately 200 RU of antibody was captured onto a Protein A Series S Sensor Chip (Cytiva) along with 200 RU of an influenza mAb AG7C in a reference flow cell. Multi cycle kinetic analysis of binding was undertaken, using a two-fold serial dilution of RBD-H in

HBS-P+ buffer (Cytiva) along with a reference sample containing only HBS-P+ buffer. Measurements were made with injection times of 90 s (30 μl/min) and dissociation times of 180 s or 600 s (30 μl/ml) at 37 °C. Regeneration of the sensor chip was performed with 10 mM Glycine-HCl, pH 1.7 for 30s (30 μl/min) between RBD-H concentrations. For analysis, the sensograms were double reference subtracted and fitted with a 1:1 binding model using the BIAcore Insight Evaluation Software version 2.0.15.12933 (Cytiva).

**Monovalent SPR for CD19 antibodies.** Monovalent SPR of CD19 antibodies was conducted on BIAcore T200 instrument (GE Healthcare Life Sciences). The experiment was run at 37 °C, HBS-EP was used as running buffer. An Fc capture chip was produced by amine coupling of anti-mouse IgG antibody to a CM5 chip using a commercial Fc Capture Kit (GE Healthcare). First, the chip was conditioned with 7 conditioning cycles using HBS-EP buffer. Next, the anti-CD19 antibody was injected for either 120 s or 900 s, resulting in immobilisation levels of 500 RU and 1000 RU respectively. For the control flow cell, the BBM1 antibody was immobilised to matching levels. CD19-SpyCatcher was injected for 150 s, followed by a dissociation phase of 450 s at a flow rate of 50 μl/min. After each cycle, the chip was regenerated with 10 mM Glycine-HCl, pH1.7 for 90 s, which removed both CD19 and anti-CD19 antibody from the chip. This was followed by the re-immobilisation of anti-CD19 antibody. Buffer was injected after every second cycle. For data analysis, the SPR sensogram was double-referenced against an empty flow cell and buffer injections. Subsequently, we fitted the dissociation phase with a 1:1 binding model to obtain the mean $k_{off}$ value. Next, the mean $k_{on}$ value was determined by fitting the association phase with a 1:1 binding model, while constraining the $k_{off}$ parameter to the mean $k_{off}$ value determined in the first fitting step.

**Bivalent SPR.** A BIAcore T200 instrument (GE Healthcare Life Sciences) at 37 °C and a flow rate of 100 μl/min. Running buffer was HBS-EP. Monovalent Streptavidin-SpyCatcher was coupled to CM5 sensor chips using an amino coupling kit (GE Healthcare Life Sciences) to near saturation, typically around 7000–8000 response units (RU). Antigens, either biotinylated (RBD) or containing a SpyTag (CD19), were injected into the experimental flow cells (FCs) for different lengths of time to produce desired immobilisation levels (typically 20–70 RU). The concentration of immobilised antigen was calculated using an empirical factor to convert the immobilisation level to a molar concentration. We used the formula: molar conc. = immobilisation level / (conversion factor × molecular weight). The conversion factor was previously determined to be 149 RU per g/liter[36]. Usually, FC1 was kept blank as a reference for FC2, FC3, and FC4. Excess streptavidin was blocked with two 40 s injections of 250 μM biotin (Avidity). Before antibody injections, the chip surface was conditioned with 2 injections of the running buffer. Dilution series of antibodies was injected simultaneously in all FCs, starting with the lowest concentration. Antibodies were injected for 150s followed by a buffer injection of 450s at a flow rate of 100 μl/min. After each cycle, the chip surface is regenerated with 3M MgCl (Cytiva) for 90 s at 30 μl/min to remove all remaining bound antibodies. A buffer injection was included after every 2 or 3 antibody injections; all binding data were double-referenced by subtracting the response of the control flow cell and the closest buffer injection. Before running the high-throughput SPR experiment, we conducted an initial screen to determine whether the surface could be regenerated after each antibody injection. For this, each antibody was injected at a concentration of 50 nM with an association phase of 150 sec followed by a dissociation phase of 475 sec before injecting 3M MgCl (Cytiva) for 90 sec at a flow rate of 30 μl/min followed by a buffer injection. For the high-throughput SPR experiments, up to 32 antibodies at 3 concentrations (typically between 30 and 100 nM) each were injected in sequence over the SPR chip surface. Buffer was injected after every third cycle.

## ODE-based monovalent model

As illustrated in Fig. 1a, the monovalent ODE model assumes soluble antigen can reversibly bind to immobilised antibodies. Let [Ag] denote the concentration of antigen, [Ab] the concentration of antibodies immobilised on the SPR chip, $A$ the number of unbound antibody arms, and $B$ the number of bound antibody arms. The monovalent reaction model is then

$$A \underset{k_{off}}{\overset{k_{on}[Ag]}{\rightleftharpoons}} B \qquad (1)$$

with the corresponding mass action ODE model of

$$\frac{dA}{dt} = -\frac{dB}{dt} = -k_{on}[Ag]A + k_{off}B. \qquad (2)$$

All antibody arms are initially unbound, giving the initial condition that $A(0) = 2[Ab]$ and $B(0) = 0$.

The monovalent SPR experiment is modelled by an association phase ($t = 0$ to $t = t_s$) followed by a dissociation phase after the instantaneous removal of antigen in solution at time $t_s$ (i.e., setting [Ag] = 0 at $t_s$). The measured monovalent SPR response trace, $R(t)$, is proportional to $B(t)$. Analytically solving the ODE model, we then have that

$$R(t) = \begin{cases} C_p \frac{2k_{on}[Ag][Ab]}{k_{on}[Ag]+k_{off}} \left(1 - e^{-(k_{on}[Ag]+k_{off})t}\right), & t \le t_s, \\ R(t_s^-)e^{-k_{off}(t-t_s)}, & t > t_s, \end{cases} \qquad (3)$$

where $C_p$ denotes the constant of proportionality between $R(t)$ (in units of RU) and $B(t)$ (in units of concentration). The monovalent model for $R(t)$ then has three unknown parameters to fit, $C_p$, $k_{on}$, and $k_{off}$.

## ODE-based bivalent model

As illustrated in Fig. 1a, the bivalent ODE model assumes soluble antibodies can reversibly bind to immobilised antigens. We let [Ab] denote the concentration of antibodies in the solution, [Ag] the concentration of antigen immobilised on the SPR chip, $A$ the concentration of these antigens that are not bound to an antibody, $B$ the concentration of antibody-antigen complexes in which the antibody has one arm bound (i.e., is "singly-bound"), and $C$ the concentration of antibodies with both arms bound to antigen (i.e., "doubly-bound"). The well-mixed bivalent reaction model is then

$$A \underset{k_{off}}{\overset{2k_{on}[Ab]}{\rightleftharpoons}} B \qquad (4)$$

$$A + B \underset{2k_{off}}{\overset{k_{on,b}}{\rightleftharpoons}} C \qquad (5)$$

with the corresponding mass action ODE model of

$$\frac{dA}{dt} = -2k_{on}[Ab]A + k_{off}B - k_{on,b}AB + 2k_{off}C \qquad (6)$$

$$\frac{dB}{dt} = 2k_{on}[Ab]A - k_{off}B - k_{on,b}AB + 2k_{off}C \qquad (7)$$

$$\frac{dC}{dt} = k_{on,b}AB - 2k_{off}C, \qquad (8)$$

and the initial conditions that $A(0) = [Ag]$, $B(0) = 0$, $C(0) = 0$. Note that here $k_{on}$ and $k_{off}$ represent the same physical rates used in the monovalent ODE model.

As with the monovalent SPR experiments, the bivalent SPR experiment is modelled by an association phase ($t = 0$ to $t = t_s$) followed by a dissociation phase after the instantaneous removal of antibodies in solution at time $t_s$ (i.e., setting [Ab] = 0 at $t_s$). The measured SPR response, $R(t)$, is proportional to the amount of antibodies bound to immobilised antigen, and we assume $R(t) = C_p(B(t) + C(t))$. We then obtain a final model with four parameters to fit, $k_{on}$, $k_{off}$, $k_{on,b}$, and a constant of proportionality $C_p$. Note that this model assumes that a singly-bound antibody can bind any free antigen on the surface (i.e., is 'well-mixed') and therefore, does not contain a molecular reach parameter. The model was fit to bivalent SPR data for FD-11A binding RBD (Fig. 1a–c) using *lsqcurvefit* in Matlab (Mathworks, MA).

## Particle-based bivalent model

The particle-based model modifies the bivalent ODE model by explicitly resolving the position and chemical state (i.e., free or antibody-bound) of each individual antigen that is immobilised on the SPR chip. We model a small portion of the SPR chip by a cube with side lengths $L$ containing $N_{Ag}$ antigens that are uniformly (randomly) distributed.

Our model is given in terms of stochastic jump processes for the states of each individual antigen or antigen-antibody complex. Let $x_i$ denote the position of the $i$th antigen within the domain, $i = 1, \ldots, N_{Ag}$. We denote by $A_i(t) \in \{0, 1\}$ the stochastic process that is one if the $i$th antigen is not bound to any antibody, and zero otherwise. Similarly, $B_i(t) \in \{0, 1\}$ will denote the stochastic process that is one if the $i$th antigen is bound to an arm of an antibody for which the other arm is unbound, and zero otherwise. Finally, $C_{ij}(t) \in \{0, 1\}$ will denote the stochastic process that is one if the antigens at $x_i$ and $x_j$ are both bound to the same antibody, and zero otherwise.

We let $\varepsilon$ label the reach of the reaction for the free arm of a singly-bound antibody-antigen complex to bind a nearby free antigen. We assume the reaction can occur with rate $k_{on,b}$ when the two antigens involved in the reaction are separated by less than $\varepsilon$. For antigens at $x_i$ and $x_j$, the rate of the reaction is given by a Doi interaction model[37–39] as $k_{on,b} \mathbb{1}_{[0,\varepsilon]}(x_i - x_j)$, where

$$\mathbb{1}_{[0,\varepsilon]}(x_i - x_j) = \begin{cases} 1, & |x_i - x_j|_p \leq \varepsilon, \\ 0, & |x_i - x_j|_p > \varepsilon \end{cases} \quad (9)$$

represents the indicator function of the interval $[0, \varepsilon]$, and $|x_i - x_j|_p$ represents the periodic distance between $x_i$ and $x_j$.

Our overall reaction model is then

$$
\begin{aligned}
A_i &\underset{k_{off}}{\overset{2k_{on}[Ab]}{\rightleftharpoons}} B_i, & i &\in \{1, \ldots, N_{Ag}\} \\
A_i + B_j &\xrightarrow{k_{on,b} \mathbb{1}_{[0,\varepsilon]}(x_i - x_j)} C_{\min\{i,j\}, \max\{i,j\}}, & i &\in \{1, \ldots, N_{Ag}\}, j \in \{1, \ldots, i-1, i+1, \ldots, N_{Ag}\}, \\
C_{ij} &\xrightarrow{k_{off}} A_i + B_j, & i &\in \{1, \ldots, N_{Ag}\}, j \in \{i+1, \ldots, N_{Ag}\}, \\
C_{ij} &\xrightarrow{k_{off}} A_j + B_i, & i &\in \{1, \ldots, N_{Ag}\}, j \in \{i+1, \ldots, N_{Ag}\},
\end{aligned}
\quad (10)
$$

where $k_{on}$ and $k_{off}$ should represent the same rates as in the monovalent and bivalent ODE models. The initial condition for each stochastic process is then $A_i(0) = 1$ for all $i$, $B_i(0) = 0$ for all $i$, and $C_{ij}(0) = 0$ for all $i$ and $j$.

The corresponding mathematical model for the evolution of the stochastic jump processes is given by Kurtz's time-change representation[40,41]. Equivalently, the probability the processes are in a given state can be described by the Chemical Master Equation (CME). Let $\mathcal{I} = \{(i,j) \mid i \neq j, i = 1, \ldots, N_{Ag}, j = 1, \ldots, N_{Ag}\}$ denote the indices of all possible $A_i$ and $B_j$ pairs, and $\hat{\mathcal{I}} = \{(i,j) \mid i = 1, \ldots, N_{Ag}, j = i+1, \ldots, N_{Ag}\}$ denote the indices of all distinct antigen pairs. For the time-change representation each possible reaction is associated with a unit rate Poisson counting process, labelled by $\{Y_{1,i}(t)\}_{i=1}^{N_{Ag}}$, $\{Y_{2,i}(t)\}_{i=1}^{N_{Ag}}$,

$\{Y_{3,i,j}(t)\}_{(i,j) \in \mathcal{I}}$, and $\{Y_{4,i,j}(t)\}_{(i,j) \in \mathcal{I}}$. We can then represent the stochastic processes for the total number of occurrences of the $A_i \to B_i$, $B_i \to A_i$, $A_i + B_j \to C_{\min\{i,j\}, \max\{i,j\}}$, and $C_{\min\{i,j\}, \max\{i,j\}} \to A_i + B_j$ reactions respectively as

$$\mathcal{N}_{1,i}(t) = Y_{1,i}\left(2k_{on}[Ab]\int_0^t A_i(s^-)\, ds\right), \; i \in \{1, \ldots, N_{Ag}\} \quad (11)$$

$$\mathcal{N}_{2,i}(t) = Y_{2,i}\left(k_{off}\int_0^t B_i(s^-)\, ds\right), \; i \in \{1, \ldots, N_{Ag}\} \quad (12)$$

$$\mathcal{N}_{3,i,j}(t) = Y_{3,i,j}\left(k_{on,b}\mathbb{1}_{[0,\varepsilon]}(x_i - x_j)\int_0^t A_i(s^-)B_j(s^-)\, ds\right), \; (i,j) \in \mathcal{I} \quad (13)$$

$$\mathcal{N}_{4,i,j}(t) = Y_{4,i,j}\left(k_{off}\int_0^t C_{\min\{i,j\}, \max\{i,j\}}(s^-)\, ds\right), \; (i,j) \in \mathcal{I}. \quad (14)$$

Our particle model is then given by

$$A_i(t) = 1 - \mathcal{N}_{1,i}(t) + \mathcal{N}_{2,i}(t) - \sum_{j \neq i} \mathcal{N}_{3,i,j}(t) + \sum_{j \neq i} \mathcal{N}_{4,i,j}(t),$$
$$i \in \{1, \ldots, N_{Ag}\} \quad (15)$$

$$B_i(t) = \mathcal{N}_{1,i}(t) - \mathcal{N}_{2,i}(t) - \sum_{j \neq i} \mathcal{N}_{3,j,i}(t) + \sum_{j \neq i} \mathcal{N}_{4,j,i}(t), \; i \in \{1, \ldots, N_{Ag}\} \quad (16)$$

$$C_{ij}(t) = \mathcal{N}_{3,i,j}(t) + \mathcal{N}_{3,j,i}(t) - \mathcal{N}_{4,i,j}(t) - \mathcal{N}_{4,j,i}(t), \; (i,j) \in \hat{\mathcal{I}}. \quad (17)$$

As with the ODE models, the particle model includes an association phase from $t = 0$ to $t = t_s$, followed by a dissociation phase after the assumed instantaneous removal of all antibodies in solution at the switching time $t_s$ (i.e., setting [Ab] = 0 at $t_s$). The measured SPR response, $R(t)$, is proportional to the average number of antibodies bound to immobilised antigen. That is, let

$$B(t) = \sum_{j=1}^{N_{Ag}} B_j(t), \; C(t) = \sum_{(i,j) \in \hat{\mathcal{I}}} C_{ij}(t) \quad (18)$$

be the stochastic processes for the number of singly- and doubly bound antibodies in the system at time $t$. Denoting averages by $\mathbb{E}[\cdot]$, we assume that

$$R(t) = \frac{C_p}{N_{Ag}}(\mathbb{E}[B(t)] + \mathbb{E}[C(t)]), \quad (19)$$

where $C_p$ denotes the constant of proportionality converting model concentrations to experimental response units. Note, as only the antibody concentration is varied during a single bivalent SPR experiment, $N_{Ag}$ remains constant, and we are simply defining a re-scaled constant of proportionality compared to the ODE cases. The resulting model then has five unknown parameters to fit, $k_{on}$, $k_{off}$, $k_{on,b}$, $\varepsilon$, and $C_p$.

Exact realisations of the system state, i.e., $\left(\{A_i(t)\}_{i=1}^{N_{Ag}}, \{B_i(t)\}_{i=1}^{N_{Ag}}, \{C_{ij}(t)\}_{(i,j) \in \hat{\mathcal{I}}}\right)$, can be generated by any of the many Stochastic Simulation Algorithms (SSAs)[17,42] (also known as Gillespie methods, Kinetic Monte Carlo methods, or Doob's method). For all forward simulations, we use an optimised implementation of the Gibson-Bruck Next Reaction Method (NRM) SSA[43].

**Particle-model surrogate.** While SSA simulations of the particle model (i.e., $R(t)$) can be directly fit to bivalent SPR data, we found their computational expense to be the bottleneck in our experimental workflows. To avoid parameter estimation becoming a rate-limiting step, we developed a surrogate model that approximated the particle model but allowed for rapid data fitting and hence, parameter estimation.

We first note that from the perspective of the particle model, the antibody solution concentration, [Ab], only enters in setting an effective transition rate, i.e., probability per time, of $\hat{k}_{on} \equiv 2k_{on}[Ab]$ for an individual $A_i$ to become a $B_i$. As such, in the surrogate model, we treat this as one effective parameter. We then denote by

$$\theta = (\log_{10}(\hat{k}_{on}), \log_{10}(k_{off}), \log_{10}(k_{on,b}), \varepsilon, \log_{10}(C_p)) \quad (20)$$

the vector of particle-model parameters to estimate from SPR traces, where we have log-transformed the transition rates for each possible reaction.

Let

$$R_p(t;\theta) = \frac{C_p}{N_{Ag}} (\mathbb{E}[B(t)] + \mathbb{E}[C(t)]) \quad (21)$$

label the particle model's predicted SPR trace for a given set of parameters. The surrogate model is a function of $t$ and $\theta$ defined by

$$R_s(t;\theta) = C_p S(t, \theta_1, \theta_2, \theta_3, \theta_4), \quad (22)$$

with $S(t, \theta_1, \theta_2, \theta_3, \theta_4)$ representing the surrogate's predicated SPR response for $C_p = 1$. $S$ is given by a five-dimensional linear interpolation table over $(t, \theta_1, \theta_2, \theta_3, \theta_4)$. For all surrogates in this work, we fixed $N_{Ag} = 1000$ and $[Ag_{sur}] = 125 \mu M$. The domain length was then chosen to be consistent with this density of antigen, i.e., for $N_A$ denoting Avogadro's number, $L$ satisfied

$$\frac{N_{Ag}}{L^3} = [Ag_{sur}] \times N_A \times 10^{-30} \frac{mol}{(nm)^3 \mu M}. \quad (23)$$

To construct the surrogate, $(\theta_1, \theta_2, \theta_3, \theta_4)$ were each varied over a range of values $[\underline{\theta_i}, \bar{\theta_i}]$, with samples of each $\theta_i$ uniformly spaced. That is, if parameter $\theta_i$ was sampled at $M_i$ values, they were chosen as

$$\theta_i^j = \underline{\theta_i} + j\Delta\theta_i, \quad j = 0, \dots, M_i - 1, \quad (24)$$

with $\Delta\theta_i = (\bar{\theta_i} - \underline{\theta_i})/(M_i - 1)$. The sampled reaction transition rates were then uniformly spaced in log space. We similarly partitioned time such that $t \in [0, T]$ where $T$ determined the time interval over which to fit the SPR data. We again used uniform spacing in time, $t_j = j\Delta t$, for $\Delta t = T/M_t$. For the surrogate used in this work we chose $(M_1, M_2, M_3, M_4) = (42, 40, 30, 30)$, $\Delta t = 1s$, $T = 600s$, and $M_t = 600$. $t_s$ was chosen to be 150s, consistent with the SPR protocol.

With $C_p = 1$, for each possible combination of parameters, $\theta = (\theta_1^{j_1}, \theta_2^{j_2}, \theta_3^{j_3}, \theta_4^{j_4}, 1)$, we then saved the values

$$S(t_k, \theta_1^{j_1}, \theta_2^{j_2}, \theta_3^{j_3}, \theta_4^{j_4}) = R_p(t_k;\theta), \quad k = 0, \dots, M_t \quad (25)$$

in a five-dimensional lookup table. Estimating $R_p(t_k; \theta)$ requires averaging SSA simulations of the particle model. Let $B^n(t)$ and $C^n(t)$ denote the sampled values of $B(t)$ and $C(t)$ in the $n$th simulation, $n = 1, \dots, N_{sims}$, with

$$R_p^n(t_k;\theta) = \frac{1}{N_{Ag}} (B^n(t_k) + C^n(t_k)) \quad (26)$$

the corresponding response. We then approximated

$$R_p(t_k;\theta) \approx \frac{1}{N_{sims}} \sum_{n=1}^{N_{sims}} R_p^n(t_k;\theta). \quad (27)$$

Our general protocol was to average over at least $N_{sims} = 15$ SSA samples, continuing to add SSA samples until either the estimated standard error of the samples, $\{R_p^n(t_k;\theta)\}_{n=1}^{N_{sims}}$, was below 0.01 of the sample mean at all $t_k$, or $N_{sims} = 250$ SSA samples were reached.

Given the table $\{S(t_k, \theta_1^{j_1}, \theta_2^{j_2}, \theta_3^{j_3}, \theta_4^{j_4})\}_{(k,j_1,j_2,j_3,j_4)=(1,1,1,1,1)}^{(M_t+1,M_1,M_2,M_3,M_4)}$, $S(t, \theta_1, \theta_2, \theta_3, \theta_4)$ for $t \in (0, T)$ and all $\theta_j \in (\underline{\theta_j}, \bar{\theta_j})$ could then be evaluated by linear interpolation of the bracketing tabulated values. In practice, we evaluated the surrogate at general time and parameter values using the `BSpline(Linear())` option from the Interpolations.jl library[44]. For the sizes we used, $(M_t + 1, M_1, M_2, M_3, M_4) = (601, 42, 40, 30, 30)$, the table contained model responses for 1,512,000 parameter combinations, and required approximately 7.3GB of memory to store. Surrogates were generally constructed in a few hours using 500–2000 cores on the Boston University Shared Computing Cluster.

**Particle-model data fitting.** Estimates for the log-transformed parameters $\theta$, defined in (20), were generated via minimisation of the squared error between surrogate predictions and experimental SPR responses for varying levels of [Ab]. Suppose $I$ SPR traces are being simultaneously fit (e.g., traces with different antibody concentrations injected over the same surface), with $R^{(i)}(t)$ labelling the $i$th SPR response trace for an experiment with antibody concentration $[Ab^{(i)}]$. We assume a fixed and known antigen concentration, [Ag], when simultaneously fitting multiple SPR traces generated using different antibody concentrations injected over the same surface. The experimental traces are ordered such that

$$[Ab^{(1)}] \leq [Ab^{(2)}] \leq \cdots \leq [Ab^{(I)}]. \quad (28)$$

Finally, given a current estimate for $\theta$ arising during optimisation, we define

$$\theta^{(i)} \equiv \left( \theta_1 + \log_{10}\left(\frac{[Ab^{(i)}]}{[Ab^{(1)}]}\right), \theta_2, \theta_3, \theta_4, \theta_5 \right) \quad (29)$$

$$= \left( \log_{10}\left(\hat{k}_{on}\frac{[Ab^{(i)}]}{[Ab^{(1)}]}\right), \log_{10}(k_{off}), \log_{10}(k_{on,b}), \varepsilon, \log_{10}(C_p) \right). \quad (30)$$

The overall loss function we then minimised was

$$L(\theta) = \sum_{i=1}^{I} \sum_{k=0}^{M_t} \left( R^{(i)}(t_k) - R_s(t_k;\theta^{(i)}) \right)^2, \quad (31)$$

where $R_s(t; \theta)$ is the surrogate response defined by (22). In practice we minimised this loss using the XNES natural evolution optimizer from BlackBoxOptim.jl via the Optimization.jl meta-package[45,46]. All optimisation-related parameters were left at their default values except the maximum number of iterations, which was increased to 5000. As XNES is a stochastic optimizer, we generally applied it several times and selected the estimated parameter set across all runs having the minimal loss as the consensus estimate. For more details, see the section on "Data analysis for high-throughput bivalent SPR". Fitting a single SPR dataset (multiple concentrations of a single antibody over a single antigen density) required approximately 10 min on a standard laptop.

As described in the previous section, the surrogate was constructed for a fixed concentration of antigen. To avoid producing a

new surrogate for each experiment, we conjectured that once our particle-model system was of sufficient size (i.e., a sufficiently large number of antigens for a fixed antigen concentration), the antigen concentration effectively set the average number of antigens that were within reach. In this way, we could fit SPR data generated with any antigen concentration using a single surrogate (produced with a single antigen concentration) but would need to transform the fitted reach based on the experimental antigen concentration. In other words, the biophysical reach ($\varepsilon_{phys}$) can be calculated from the fitted reach ($\varepsilon_{sur} = \theta_4$) by enforcing that the average number of antigens within reach in the surrogate model and in the SPR experiment are the same,

$$\frac{4}{3}\pi \varepsilon_{sur}^3 [\mathrm{Ag_{sur}}] = \frac{4}{3}\pi \varepsilon_{phys}^3 [\mathrm{Ag}]. \tag{32}$$

We empirically confirmed that the model produced the same predicted SPR traces at different antigen densities ([Ag] = 1, 10, and 100 μM) provided that the physical molecular reach ($\varepsilon_{phys}$) was decreased according to the above equation (Fig. S1). Therefore, the relationship between the fitted parameters $\theta$ and the biophysical parameters are as follows,

$$k_{on} = \frac{10^{\theta_1}}{[\mathrm{Ab}^{(1)}]} = \frac{\hat{k}_{on}}{[\mathrm{Ab}^{(1)}]}, k_{off} = 10^{\theta_2}, k_{on,b} = 10^{\theta_3},$$
$$\varepsilon_{phys} = \theta_4 \left(\frac{[\mathrm{Ag_{sur}}]}{[\mathrm{Ag}]}\right)^{1/3}, C_p = 10^{\theta_5}. \tag{33}$$

These represent the final biophysical parameter estimates reported in this work.

**Particle-model antibody binding potency predictions.** To predict the concentration of antibody required to bind 50% of antigen (antibody binding potency), the particle model is simulated using the previously mentioned Next Reaction Method-based approach. The only modification to the model is that we used a 2D square with sides of length $L$ to represent the 2D viral surface. As before, antigens are assumed to be uniformly (randomly) distributed in a 2D region with $N_{Ag}$ set to 1000 and $L$ chosen to enforce a specified antigen density.

For each antibody, the model was simulated with its fitted binding parameters ($k_{on}$, $k_{off}$, $k_{on,b}$, and the molecular reach $\varepsilon$) for different antigen and antibody concentrations to a time of 60 min. For each antibody concentration we averaged 20 independent simulations to estimate the average number of unbound antigens at 60 min. The resulting dose-response curves for the fraction of unbound antigen (Ag$_{unbound}$) versus antibody concentration ([Ab]) were fit with an inhibitory Hill model ($n < 0$) to determine IC$_{50}$,

$$\mathrm{Ag_{unbound}} = \frac{1}{1 + \left(\frac{\mathrm{IC_{50}}}{[\mathrm{Ab}]}\right)^n}. \tag{34}$$

**Worm-like-chain (WLC) model to estimate size of PEG-coupled antigen**

The worm-like chain (WLC) is a widely used polymer model that has previously been applied to PEG polymers[47]. The model provides an estimate for the mean end-to-end distance of the polymer as follows: $2\sqrt{N_{PEG} l_c l_p}$ where $N_{PEG}$ is the number of PEG repeats, $l_c$ is the contour length of each repeat (0.4 nm), and $l_p$ is the persistence length of PEG (0.4 nm[47]). The WLC model predicts a mean length of 0.69 nm for PEG3 and 2.1 nm for PEG28 or a difference in length of 1.41 nm. The predicted increase in molecular reach when an antibody binds PEG28 instead of PEG3 would then be twice this difference to account for the two bound antigens involved in antibody binding (2.82 nm, Fig 2C).

## Molecular dynamics

**Construction of full-length all-atom antibody/RBD complex structures.** The PDB structure (1HZH) of a full-length IgG1 antibody was rebuilt and minimised in CHARMM35[48] based on SEQRES records to serve as a template for rebuilding of full-length antibodies specific for RBD. The crystal structures of the Fab/RBD complexes used are listed in Table S1. Each RBD structure was rebuilt to contain all residues of the N-terminal AviTag and signal sequence (24 residues total after cleavage) used in SPR experiments fused to RBD residues 331–526 (196 residues) resulting in a final construct of 220 residues in all cases. Any missing residues in the light or heavy chains of the Fabs were also rebuilt and all rebuilt sections of Fabs and RBD minimised in CHARMM35. Two copies of each Fab/RBD complex were then aligned to the 1HZH all-atom model in Chimera v1.16[49] using the MatchMaker tool based on the heavy and light chain only, with the relative orientations of the RBD and Fab structures maintained. Heavy chains from these aligned structures were then merged with those of 1HZH based on MUSCLE[50] alignments of their sequences and inspection of the aligned structures. The three residues on either side of the new bond were minimised in CHARMM35 with all other atomic positions held fixed. The final structure after 1000 steps of unconstrained minimisation in vacuo in Amber FF14SB[51] with OpenMM v7.5[52] was accepted as the final all-atom structure. We note that 1HZH contains one disulfide bond between the two heavy chains in the hinge region rather than the usual two. We chose not to rebuild the second disulfide bond so as not to distort the overall structure and because the disulfide that limits the extension of the hinge is in place. The lengths of all chains within each antibody in the rebuilt full-length models are listed in Table S2.

**Construction of topology-based coarse-grain models.** Coarse-grain models were parameterised following a previously published protocol that represents each amino acid as a single interaction site centred at the $C_\alpha$ coordinates of each atom. The potential energy of a conformation within this model is given by

$$E = \sum_i k_b(r_i - r_0)^2 + \sum_i \sum_j^4 k_{\varphi_{ij}}\left(1 + \cos\left[j\varphi_i - \delta_{ij}\right]\right) + \sum_i -\frac{1}{\gamma}\ln\left\{\exp\left[-\gamma\left(k_\alpha(\theta_i - \theta_\alpha)^2 + \varepsilon_\alpha\right)\right]\right.$$
$$\left. + \exp\left[-\gamma k_\beta\left(\theta_i - \theta_\beta\right)^2\right]\right\} + \sum_{ij} \frac{q_i q_j e^2}{4\pi\varepsilon_0\varepsilon_r r_{ij}}\exp\left[-\frac{r_{ij}}{l_D}\right] + \sum_{ij\in\{NC\}} \epsilon_{ij}^{NC}\left[13\left(\frac{\sigma_{ij}}{r_{ij}}\right)^{12} - 18\left(\frac{\sigma_{ij}}{r_{ij}}\right)^{10}\right.$$
$$\left. + 4\left(\frac{\sigma_{ij}}{r_{ij}}\right)^6\right] + \sum_{ij\notin\{NC\}} \epsilon_{ij}^{NN}\left[13\left(\frac{\sigma_{ij}}{r_{ij}}\right)^{12} - 18\left(\frac{\sigma_{ij}}{r_{ij}}\right)^{10} + 4\left(\frac{\sigma_{ij}}{r_{ij}}\right)^6\right].$$
$$\tag{35}$$

These forcefield terms, which represent contributions from bonds, dihedrals, angles, electrostatics, native contacts, and non-native contacts, have been described in detail previously[53]. The strength of native contacts within this coarse-grain model is determined by the Lennard-Jones-like well depths for the native contact term. The well depth, $\epsilon_{ij}^{NC}$, is computed as $\epsilon_{ij}^{NC} = \eta_{ij}\epsilon_{HB} + \eta\epsilon_{ij}$, in which $\eta_{ij}$ is the number of hydrogen bonds between residues $i$ and $j$, $\epsilon_{HB} = 0.75$ kcal/mol, $\epsilon_{ij}$ is the well depth taken from the Betancourt-Thirumalai[54] pairwise potential for a contact between residues of types $i$ and $j$, and $\eta$ is a scaling factor that increases the effective well depths. Increasing the value of $\eta$ linearly increases the stability of all native contacts to which it is applied. Values of $\eta$ were selected individually for each domain and interface within the antibody-RBD complexes. We note that disulfide bonds are treated as harmonic bonds with $k_b = 20$ kcal/(mol × Å$^2$) rather than as native contacts.

**Selection of $\eta$ values for antibody domains and interfaces.** All coarse-grain simulations were performed in OpenMM v7.5 using the LangevinMiddleIntegrator[55] with an integration time step of 15 fs, a friction coefficient of 0.050 ps$^{-1}$, and an absolute temperature of 298 K. Values of $\eta$ were selected based on a previously published training

set[56]. To limit the number of domains and interfaces that required simultaneous parameter tuning, we used the same set of $\eta$ values for all domains and interfaces within each antibody and for each RBD monomer (Table S3). The intra-domain values of $\eta$ were taken in each case as the largest value from a training set (see ref. 57 Table S5). Likewise, the largest training set value for interfaces was used for the HC1|LC1, HC2|LC2, and HC1:HC2 interfaces. The HC1|LC2 interface, which likely represents a crystal packing interface, was assigned the training set average value of 1.507. Simulations of FD-11A in isolation were performed using the values of $\eta$ in Table S3 but with the RBD representations deleted.

**Coarse-grain steered molecular dynamics simulations of antibody molecular reach.** Initial structures for steered molecular dynamics were generated by aligning the coarse-grained RBD-antibody-RBD complexes such that one RBD Lys15 (i.e., the residue that is biotiny-lated and bound to streptavidin in the experiment) was at the coordinate system origin and the other on the positive x-axis. A spherical harmonic restraint with force constant 50 kcal/(mol × Å$^2$) was applied to the Lys15 at the origin to hold it in place during pulling. A second spherical harmonic restraint with force constant 1 kcal/(mol × Å$^2$) was applied to the second Lys15 residues to serve as the trap for steered molecular dynamics. Flat-bottom Root Mean Square Deviation (RMSD) restraints with force constant $k_{RMSD} = 50$ kcal/(mol × Å$^2$) and RMSD threshold of 5 Å were applied to residues in Fab1, Fab2, the Fc, as well as the globular portions of each RBD monomer to prevent overall unfolding while allowing structural fluctuations. Only the RBD linkers, antibody hinge, and epitope|paratope interfaces were left unrestrained. Each trajectory was then equilibrated for 1.5 ns to allow the coordinates to randomise. After equilibration, the harmonic restraint on the second Lys15 residue was pulled with a constant velocity of 1 nm/ns along the positive x-axis to a total displacement of +30 nm from its original position over the course of 30 ns. A total of 50 statistically independent trajectories were run for each of the six antibodies and each of seven different epitope|paratope interface $\eta$ values, $\eta = 1.000, 1.250, 1.500, 1.750, 2.000, 2.250, 2.500$ for a total of 2100 statistically independent trajectories (total simulation time 66 μs). The molecular reach was computed from each trajectory as longest distance at which both epitope|paratope interfaces have a non-zero fraction of native contacts, Q. Larger values of $\eta$ sometimes result in unfolding of RBD or IgG domains during pulling due to the increased interaction strength at the epitope|paratope interfaces. We therefore applied a filter to discard trajectories that ever have a Fab or RBD RMSD > 10 Å. Varying this threshold value to 8 or 15 Å was not found to strongly influence results. The mean maximum Lys15-Lys15 distance was then computed by averaging the value of those trajectories that remain reasonably well folded. The molecular reach estimate from the simulations was taken to be the maximum mean value across all $\eta$ values.

A sample of the MD simulations can be found in figshare[58].

**SARS-CoV-2 neutralisation**
Neutralisation IC$_{50}$ for the FD-11A antibody was measured with a microneutralization assay as previously described[59]. Briefly, two doses of SARS-CoV-2 vaccination induce robust immune responses to emerging SARS-CoV-2 variants of concern. FD-11A was preincubated with SARS-CoV-2 for 60 min at room temperature before being added to Vero CCL-81 cells. Level of infection was measured by counting the number of infectious foci.

For anti-RBD antibodies used in the high-throughput SPR experiment, neutralisation IC$_{50}$ was measured using a Focus Reduction Neutralisation Test as described in ref. 5. Briefly, serially diluted antibody was incubated with SARS-CoV-2 for 1 h at 37 °C, afterwards transfered to Vero cell monolayers. The level of infection was measured using a focus forming assay.

**Data analysis for high-throughput bivalent SPR**
In this section, we explain the detailed workflow for analysis of the high-throughput bivalent SPR experiments (Fig. 4).

After double referencing the SPR curves for each antibody concentration, we aligned them to the start of the dissociation phase (setting it to $t_s = 150$ s) to improve curve alignment. To eliminate artefacts arising from the start and end of the association phase (generally large spikes in RU arising from needle motion), the first 5 s of the association phase, the last 4 s of the association phase, and the first 5 s of the dissociation phase were excluded from the data. We also excluded all SPR curves where the maximum response across the entire injection was smaller than 6 RU because this minimal binding is within the systematic error for SPR experiments (typically 1 RU per 100 s of injection or 6 RU for our 600 s experiments). Finally, to produce more manageable file sizes, we reduced the temporal resolution from 10 Hz to 1 Hz and this did not impact our results because the kinetics of the SPR traces were much slower than 1 Hz.

We then fitted the processed data for each antibody using the bivalent Particle-based model (see "Particle-model data fitting" section). The antigen and antibody concentrations for each SPR curve were provided as input parameters for the fitting process. The surrogate model used in fitting contained the following parameter ranges: $\log_{10}(k_{on}) \in [-5.0, 2.0]$ with $M_1 = 42$, $\log_{10}(k_{off}) \in [-4.0, 0.0]$ with $M_2 = 40$, $\log_{10}(k_{on,b}) \in [-3.0, 1.5]$ with $M_3 = 30$, and $\varepsilon \in [2, 35]$ with $M_4 = 30$. During fitting of the surrogate to SPR data, a box constraint that $\log_{10}(C_p) \in [1.0, 4.0]$ was used. The fitting process was repeated 100 times, and the parameters yielding the lowest fitness were recorded. The estimated bivalent model parameters from the fitting process were converted into the corresponding biophysical parameters according to (33) using the experimental antigen concentration. The bivalent SPR data was also fit with the ODE-based monovalent model (see "ODE-based monovalent model" section).

Finally, a quality control procedure was introduced to ensure the accuracy of the bivalent binding parameters. First, we checked that the antigen surface could be regenerated after each antibody injection (7 out of 80 antibodies could not be removed). Second, we only included data where the particle model produces a close fit to the data (12 out of 80 antibodies could not be fit). Third, we only included data where the particle model produced a fit that was better than the ODE-based monovalent model. We reasoned that SPR data that could accurately be fit by the ODE-based monovalent model did not contain information that could accurately determine bivalent binding (bivalent binding parameters of 16 out of 80 antibodies could not be determined).

Bivalent SPR data that passed all quality control measures for each antibody were averaged across SPR experiments. We report the geometric mean for parameters that varied on a logarithmic range in the model ($k_{on}$, $k_{off}$, $K_D$, $k_{on,b}$) and the mean for the molecular reach that varied on a linear range in the model.

**Statistical analysis**
**General statistical analysis.** Statistical analyses were conducted using GraphPad Prism (v 9.5.1). Data comparisons were made with two-sided $t$-tests, with corrections for multiple comparisons using either the Holm-Šídák or Dunnett methods, as appropriate. An F-test was performed to assess whether two models differed significantly, specifically to evaluate if a linear fit diverged from the line of identity. A Pearson correlation matrix was calculated to explore correlations between binding parameter variables. Single and multiple linear regressions were performed to examine correlations between antibody binding parameters and antibody function, with the $R^2$ value reported to indicate the strength of each association. Statistical significance was set at $P < 0.05$.

**Multiple linear regression.** We fit a one-way multiple linear model in GraphPad Prism (v 9.5.1) with the following formula:

$$y = \beta_0 + \beta_1 x_1 + \ldots + \beta_n x_n \qquad (36)$$

where $y$ is the predicted neutralisation $IC_{50}$ for each antibody, $\beta_0$ is the y-intercept to fit, $(x_1, \ldots, x_n)$ are the values for $\log_{10}(k_{on})$, $\log_{10}(k_{off})$, $\log_{10}(K_D)$, $\log_{10}(k_{on,b})$, $\varepsilon$, and the blocking epitope distance respectively, and $(\beta_1, \ldots, \beta_n)$ are the corresponding regression coefficients for each variable. Models contain either all variables or a subset. A Least Squares regression type was used, and models were compared using an Extra-Sum-of-Squares F test.

**Spike density calculation.** Ke et al.[22] published estimates of the average diameter of SARS-CoV-2 virions ($\phi = 91 \pm 11$ nm) and the average number of Spike molecules per virion ($N = 24 \pm 9$), obtained from cryo-electron tomography images. Data are presented as mean ± SD. We calculated the density of Spike molecules, $\rho$, by dividing the number of molecules by the virion's surface area, assuming the virion is spherical:

$$\rho = \frac{N}{\pi \phi^2} \qquad (37)$$

The error in $\rho$ was calculated using error propagation, as follows:

$$\frac{\delta \rho}{\rho} = \sqrt{\left(\frac{\delta N}{N}\right)^2 + \left(\frac{\delta \phi}{\phi}\right)^2} \qquad (38)$$

### Reporting summary
Further information on research design is available in the Nature Portfolio Reporting Summary linked to this article.

## Data availability
Source data containing data used to make each main figure in the manuscript are provided in the Supplementary Information/Source Data file. The file also contains a spreadsheet summarising the fitted parameters, neutralisation potency, epitope, predicted binding potency at different densities, and the exclusion/inclusion criteria for each antibody. A file containing all of the referenced and aligned SPR traces for every antibody experiment in the manuscript can be found on figshare[60]. This data can be directly fitted with the particle-based model to reproduce the fits. A sample of the MD simulations can also be found in figshare[58]. Source data are provided with this paper.

## Code availability
Codes and tutorials for particle-model forward simulation, surrogate construction, and fitting are available in our MIT-licensed Julia library[61].

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

## Acknowledgements

We thank Citlali Solis Salas, Jack Paget, and Philip K. Maini for their effort to formulate and analyse a master equation representation of the particle-model. We thank Lisa Schimanski and Pramila Rijal for their assistance in producing monoclonal antibodies. Surrogate construction was carried out using the Shared Computing Cluster administered by Boston University's Research Computing Services. The work received funding from a Wellcome Trust Senior Fellowship in Basic Biomedical Sciences (207537/Z/17/Z to O.D.) and the National Science Foundation (NSF-DMS 1902854 and NSF-DMS 2325185 to S.A.I., supporting S.A.I., D.B.W., and Y.Z.). We acknowledge the NIH Research Biomedical Research Centre Funding Scheme (to G.R.S.) and the Chinese Academy of Medical Sciences Innovation Fund for Medical Science, China (2018-I2M-2-002 to D.I.S./G.R.S.). We are also grateful for a Fast Grant (Fast Grants, Mercatus Center), Schmidt Futures, and the Red Avenue Foundation. The Wellcome Centre for Human Genetics is supported by the Wellcome Trust (090532/Z/09/Z).

## Author contributions

Conceptualisation: O.D., S.A.I. Methodology: A.H., D.B.W., D.N., M.A.K., O.D., S.A.I. Software: A.H., D.B.W., D.N., S.A.I. Formal analysis: A.H., D.N. Investigation: A.H., D.B.W., D.N., M.A.K., O.D., P.A.v.d.M., R.D., S.A.I., T.K.T., W.J., C.M.D. Data curation: A.H., D.N., S.A.I. Visualisation: A.H., D.N., O.D., S.A.I. Writing—original draft: A.H., D.N., O.D., S.A.I. Writing—review & editing: A.H., D.N., O.D., P.A.v.d.M., S.A.I. Resources: A.T., C.L., G.S., J.M., M.I.B., P.S., T.K.T., W.D., W.J., Y.Z. Project administration: C.M.D., O.D., S.A.I. Funding acquisition: C.M.D., O.D., S.A.I. Supervision: C.M.D., O.D., S.A.I.

## Competing interests

G.R.S. is on the GSK Vaccines Scientific Advisory Board, a founder shareholder of RQ Biotechnology, and a Jenner investigator. Oxford

University holds intellectual property related to the Oxford-AstraZeneca vaccine. The remaining authors declare no competing interests.
