## [Peer Review File · Nature Communications]

REVIEWER COMMENTS

Reviewer #1 (Remarks to the Author):

The manuscript by Huhn et al studies the impact of molecular reach, i.e. the maximal distance that can be covered by an antibody against an epitope on two copies of an antigen attached to a (relatively) fixed surface. They develop a novel particle-based model which predicts neutralization potency of a large panel of SARS-CoV2 antibodies well. The study is highly interesting and provides novel means for computing the neutralizing activity of antibodies against certain antigens and viruses. The

The experiments are very well performed, and the manuscript is clearly written. In my opinion, it will be of interest to the broad readership of Nature Communication.

I have a few suggestions for improvement.

The authors are studying the impact of avidity binding, such as reviewed in Oostindie et al, Nature Reviews Drug Discovery, 2022. Some words about the distinction between affinity and avidity might be added. It may be worth noting that additional downstream tiers of avidity binding may impact the antibody's activity in vivo, including the activation of effector functions. These might well be impacted by the principles studied here.

The authors very much focus on the importance of reach and show that reach for the protein antigens studied is greater than the maximal physical separation between the antigen binding sites on both IgG1 Fab arms. Only in the discussion it becomes clear that reach essentially is defined as the combined distance that can be spanned by the antibody Fab arms combined with the positioning and length of the antigen (e.g. lines 236-240). The abstract just mentions 'due to antigen size'. I would suggest to bring the concept of reach more to the forefront. It puzzled me somewhat and some more general readers might miss the explanation in the discussion.

Some discussion should be added that the concept for reach and its correlation with neutralization potency, as studied here, is applicable to immobilized antigens (such as on the biosensor surface or present in a relatively fixed position on a virus). Unless, I am missing something, it remains unclear if antibodies against highly mobile antigens, would demonstrate similar restrictions and correlations. Particularly high affinity antibodies, which would remain bound for extended periods of time, could benefit from antigen mobility to facilitate bivalent binding (e.g. pertaining to the high affinity antibodies that were excluded (line 139), because the particle model did not provide additional benefit over the ODE model).

The authors choose to only study IgG1 antibodies. It would be very interesting to study, at least a selection of antibodies in an IgG2, 3 or 4 subclass as well. Differences in hinge flexibility and length and the impact thereof might provide additional support and insights.

Minor: the authors mention several times (e.g. line 2) that IgG antibodies have identical binding sites and may be bivalently. This is not the case for IgG4 in human blood, which contains different binding sites

and is usually functionally monovalent.

Reviewer #2 (Remarks to the Author):

The authors investigate a critical and important current problem, that is how to evaluate the function of an antibody from its mechanistic binding affinity.

As an example, there is still no available correlate of protection for vaccines because high affinity antibodies are not necessarily neutralizing antibodies, which makes it impossible to tell if someone is protected from infection or not.

Here, the authors investigate the contribution of bivalent binding to neutralization on a real-world antigen (the covid RBD), and use a combined computational-experimental approach to the molecular reach of both arms of an antibody.

They use SPR on surfaces coated with antigens at different distances as experimental binding data, and use an ODE model for bivalent binding of the two arms, to infer back the binding capacity depending on the distance between the antigens.

The model is well suited to current good models of TCR binding, and accurately recapitulated the experimental SPR profiles. They further validated the model on antibodies with measured k_{on}/k_{off} (Fig 1f).

The authors then show the impact of different biological features that might modulate the effectiveness of binding.

First, on small antigens, they validate that the optimal antigen distance for binding is the same between predicted and measured

For a bigger antigen like the Covid RBD, they further validate their approach using molecular dynamics.

Therefore, the authors provide a validated combined approach to identify the antigen reach, which can be easily/largely used.

But the key point from the manuscript then comes from linking antibody reach and neutralization capacity. On antibodies with known affinity, epitope and neutralization capacity.

They first show that antibodies with the same affinity had a large range of antigen reach. The authors observed, as expected, that affinity doesn't predict well neutralization capacity, but that the antigen reach did correlate with neutralization capacity (and the correlation is pretty good regarding the complexity of antibody-antigen binding).

The authors then propose to test a consequence of their finding, that is that the distribution of RBD molecules (antigen density) on the virus itself will determine the distance between antigens and therefore the neutralization capacity of the antibodies.

Again, the model recapitulated well the experimental data (Figure 6c) and the authors discuss that this is only valid at intermediate densities, which is not a problem since biologically speaking the virus are not expected to use extreme amounts of antigens.

Altogether, the fact that a model can predict antigen reach and extrapolates to neutralization capacity is a big step in understanding and predicting antibody function and vaccine efficacy.

I believe this manuscript recapitulates a high amount of work (as one can see from all the supplementary figures), and the authors are honest on limitations of the model, as well as why they discarded some antibodies based on quality standards (Fig S9).

Also, there are very few papers around talking about antigen reach. There was Hoffecker 2022, which is mentioned and discussed the impact of antibodies of different reach that will preferentially move to antigens distributed with their optimal reach, but didn't extensively look at many antigens, nor neutralization at all.

There was also Amitai 2018 (<https://doi.org/10.1371/journal.pcbi.1006408>) that used a purely theoretical model to predict that there is an optimal spacing of RBDs in HIV regarding to antibody binding, but this article was pretty disappointing because the optimal was actually flat and lacked experimental evidence.

There authors also cite previous works analyzing SPR and bivalent binding, but those papers do not tackle antibodies and do not have extensive experimental validation like the present manuscript.

Altogether, I therefore believe the manuscript is original, useful and mature enough to be published. The methods are well described, and the text is pleasant to read.

Minor points:

The code is made available (thanks!) but there doesn't seem to be much documentation. An explanation of explicit experimental steps and how to use the code would be important from community to be able to use this approach (for instance showing examples in the readme files with input details and output plots).

The authors could discuss more, if they like, if the type of epitope (buried, surface, with glycans, distance from the root) could have impact on the antigen reach profile, and if their approach could indirectly inform which type of epitope was bound. Also, they could speculate on an optimal distance of antigens when using mosaic antigen vaccines with the same or different antigens and if their approach could predict in advance the efficacy of a certain mosaic design.

Reviewer #2 (Remarks on code availability):

I didn't run the code myself (no experience in Julia)
there are very few instructions on how to prepare data (modify the JI file) or interpret the results.
Examples of results in the readme would be helpful.

Reviewer #3 (Remarks to the Author):

Huhn et al propose a way to model bivalent binding of an antibody to surface bound antigens. Their

model is based on the size of the ag as well as on parameters related to paratope-epitope interactions of the single arms. They show that their models allows for better modeling of apparent affinity and correlates well with experimental results. Moreover, they show that this model also correlates well with functional results of viral neutralization.

This study offer an interesting step beyond the current established literature, and provides an interesting and useful insight on the relationships between monovalent kinetics and functional neutralization. The conclusion is supported by both the model and the experimental results for two antigens. The ms is well written and easy to follow.

Comments:

The author conclude that the reach is determined by "both the antigen and the antibody". The antigen - they show that it is specifically determine by the size of the antigen. As for the antibody - the focus of the interface with the epitope and its kinetics. I wonder, however, what role, if any antibody specific parameters such as the flexibility of the hinges, the elbow and the H-L interface may play. The mechanism that the authors expose is structural and spatial in its nature. One may expect that if an antibody is somehow more flexibly and can separate its own arms more widely, the reach will also grow. This will have ramifications for antibody engineering as well. It will be interesting to hear the authors thoughts on this aspect of the reach.

It would be interesting if the author discussed whether their observations about the correlation between the reach and neutralization potency can be generalized to other antigens (viral and others) and other types of antibodies (non-IgG1, single domain, etc)

The study's conclusions may be depend on the structure/topology of the antigen. Since it was only tested on SARS-CoV-2 RBD Abs and one CD19 Ab- the limitations of generalizing the conclusions should be further emphasized.

technical comments: In the steered MD when they pull antibody arms apart, could they estimate how much energy the steering imparts into the system? Is the estimated energy plausible under physiological conditions? How did they calculate Ab distance from the reference epitope? Are experimental ag-ab complex structures available for all tested antibodies or did they use models?

Minor comments:

Figure 1C- graphical legend appears on 1E. It may be helpful to have a table with the average and stdev values for figure (1C) and related ones in the supplementary material.

Figure S1.C the plots at different Ag concentrations seem almost identical. Consider putting them on one plot to allow comparison.

The Molecular dynamics reproduced impressively the calculated 'reach' values of antibodies that were co-crystallized with the antigens. It was important for this paper to support the discrepancy between the high calculated 'reach' values of this paper in comparison with the previous researches. The authors may want to provide representative MD files with the publication. All the files should be kept so that they may be provided upon request.

The measured correlation of the calculated 'reach' with the neutralization IC50 was good ($R^2=0.52$; $n=45$), and with all parameters (multiple linear regression) it was significantly improved ($R^2=0.82$; $n=45$). It is not clear if the linear regression was examined using cross validation. If not – I'd advise repeating it with CV.

Reviewer 1

The manuscript by Huhn et al studies the impact of molecular reach, i.e. the maximal distance that can be covered by an antibody against an epitope on two copies of an antigen attached to a (relatively) fixed surface. They develop a novel particle-based model which predicts neutralization potency of a large panel of SARS-CoV2 antibodies well. The study is highly interesting and provides novel means for computing the neutralizing activity of antibodies against certain antigens and viruses. The

The experiments are very well performed, and the manuscript is clearly written. In my opinion, it will be of interest to the broad readership of Nature Communication.

We thank the reviewer for taking the time to read our manuscript and for providing feedback.

I have a few suggestions for improvement.

The authors are studying the impact of avidity binding, such as reviewed in Oostindie et al, Nature Reviews Drug Discovery, 2022. Some words about the distinction between affinity and avidity might be added. It may be worth noting that additional downstream tiers of avidity binding may impact the antibody's activity in vivo, including the activation of effector functions. These might well be impacted by the principles studied here.

It is interesting that by the definition of Oostindie et al, molecular reach would be expected to modulate antibody avidity. We have added this point to the discussion along with a reference to the paper.

The authors very much focus on the importance of reach and show that reach for the protein antigens studied is greater than the maximal physical separation between the antigen binding sites on both IgG1 Fab arms. Only in the discussion it becomes clear that reach essentially is defined as the combined distance that can be spanned by the antibody Fab arms combined with the positioning and length of the antigen (e.g. lines 236-240). The abstract just mentions 'due to antigen size'. I would suggest to bring the concept of reach more to the forefront. It puzzled me somewhat and some more general readers might miss the explanation in the discussion.

We have now included an explicit statement in the abstract.

Some discussion should be added that the concept for reach and its correlation with neutralization potency, as studied here, is applicable to immobilized antigens (such as on the biosensor surface or present in a relatively fixed position on a virus). Unless, I am missing something, it remains unclear if antibodies against highly mobile antigens, would demonstrate similar restrictions and correlations. Particularly high affinity antibodies, which would remain bound for extended periods of time, could benefit from antigen mobility to facilitate bivalent binding (e.g. pertaining to the high affinity antibodies that were excluded (line 139), because the particle model did not provide additional benefit over the ODE model).

We have revised the discussion to include this point.

We have plotted the potency of antibodies based on the inclusion/exclusion criteria (Fig 1 below) and find that we do not have any systematic bias (i.e. it is not the case that only highly potent antibodies were removed from the analysis). As a result, our inclusion/exclusion criteria are unlikely to impact our conclusions.

The authors choose to only study IgG1 antibodies. It would be very interesting to study, at least a selection

Figure 1: Neutralisation potency of antibodies classified based on inclusion / exclusion criteria shows a relatively uniform set of antibodies were excluded (bad particle model fit or equal monovalent model fit) from the analysis (i.e. no bias can be detected in which antibodies were excluded).

Figure 2: Comparison between binding parameters for IgG1 and IgA antibody classes. Antibody FD-11A was produced as IgG1 and IgA antibody. No significant difference in binding parameters could be detected.

of antibodies in an IgG2, 3 or 4 subclass as well. Differences in hinge flexibility and length and the impact thereof might provide additional support and insights.

A previous study (Shaw et al (2019) Nature Nanotechnology; <https://www.nature.com/articles/s41565-018-0336-3>) used a DNA origami platform to study the maximum antigen separation of IgM and all IgG subclasses finding that they all tolerated a maximum separation distance of ~ 16 nm. Moreover, another study reached the same conclusion for IgE (Schneider et al (2023) ACS Nano; <https://doi.org/10.1021/acsnano.2c12647>). We therefore decided to examine the molecular reach of a different isotype, namely monomeric bivalent IgA antibodies (i.e. without the J-chain). We now find that the FD-11A antibody on the IgA rather than IgG1 hinge reproduces all of the same parameters, including molecular reach (Fig 2 below). Thus, antibody hinge does not appear to have a dramatic impact on binding properties. We have included text in the results along with a supplementary figure to explain this new result.

Minor: the authors mention several times (e.g. line 2) that IgG antibodies have identical binding sites and

may be bivalently. This is not the case for IgG4 in human blood, which contains different binding sites and is usually functionally monovalent.

We were not aware of this study and have revised line 2 to say 'usually' and included this point explicitly in the discussion along with a reference to Kofschoten et al (2007) Science.

Reviewer 2

The authors investigate a critical and important current problem, that is how to evaluate the function of an antibody from its mechanistic binding affinity. As an example, there is still no available correlate of protection for vaccines because high affinity antibodies are not necessarily neutralizing antibodies, which makes it impossible to tell if someone is protected from infection or not. Here, the authors investigate the contribution of bivalent binding to neutralization on a real-world antigen (the covid RBD), and use a combined computational-experimental approach to the molecular reach of both arms of an antibody.

They use SPR on surfaces coated with antigens at different distances as experimental binding data, and use an ODE model for bivalent binding of the two arms, to infer back the binding capacity depending on the distance between the antigens. The model is well suited to current good models of TCR binding, and accurately recapitulated the experimental SPR profiles. They further validated the model on antibodies with measured k_{on}/k_{off} (Fig 1f).

The authors then show the impact of different biological features that might modulate the effectiveness of binding. First, on small antigens, they validate that the optimal antigen distance for binding is the same between predicted and measured. For a bigger antigen like the Covid RBD, they further validate their approach using molecular dynamics. Therefore, the authors provide a validated combined approach to identify the antigen reach, which can be easily/largely used.

But the key point from the manuscript then comes from linking antibody reach and neutralization capacity. On antibodies with known affinity, epitope and neutralization capacity. They first show that antibodies with the same affinity had a large range of antigen reach. The authors observed, as expected, that affinity doesn't predict well neutralization capacity, but that the antigen reach did correlate with neutralization capacity (and the correlation is pretty good regarding the complexity of antibody-antigen binding).

The authors then propose to test a consequence of their finding, that is that the distribution of RBD molecules (antigen density) on the virus itself will determine the distance between antigens and therefore the neutralization capacity of the antibodies. Again, the model recapitulated well the experimental data (Figure 6c) and the authors discuss that this is only valid at intermediate densities, which is not a problem since biologically speaking the virus are not expected to use extreme amounts of antigens.

Altogether, the fact that a model can predict antigen reach and extrapolates to neutralization capacity is a big step in understanding and predicting antibody function and vaccine efficacy.

I believe this manuscript recapitulates a high amount of work (as one can see from all the supplementary figures), and the authors are honest on limitations of the model, as well as why they discarded some antibodies based on quality standards (Fig S9).

Also, there are very few papers around talking about antigen reach. There was Hoffecker 2022, which is mentioned and discussed the impact of antibodies of different reach that will preferentially move to antigens distributed with their optimal reach, but didn't extensively look at many antigens, nor neutralization at all. There was also Amitai 2018 (<https://doi.org/10.1371/journal.pcbi.1006408>) that used a purely theoretical model to predict that there is an optimal spacing of RBDs in HIV regarding to antibody binding, but this article was pretty disappointing because the optimal was actually flat and lacked experimental evidence. These authors also cite previous works analyzing SPR and bivalent binding, but those papers do not tackle antibodies and do not have extensive experimental validation like the present manuscript.

Altogether, I therefore believe the manuscript is original, useful and mature enough to be published. The methods are well described, and the text is pleasant to read.

We thank the reviewer for taking the time to read our manuscript and for providing feedback.

Minor points: The code is made available (thanks!) but there doesn't seem to be much documentation. An explanation of explicit experimental steps and how to use the code would be important from community to be able to use this approach (for instance showing examples in the readme files with input details and output plots).

We have now revised the GitHub repository to include documentation consisting of three tutorials on forward model simulations, building surrogates, and fitting SPR experiments; and also included a full example that can be downloaded with scripts and input data illustrating our standard workflow. A link is also provided to the full surrogate we used in the reported studies.

The authors could discuss more, if they like, if the type of epitope (buried, surface, with glycans, distance from the root) could have impact on the antigen reach profile, and if their approach could indirectly inform which type of epitope was bound. Also, they could speculate on an optimal distance of antigens when using mosaic antigen vaccines with the same or different antigens and if their approach could predict in advance the efficacy of a certain mosaic design.

We have revised the discussion to include comments on optimal spacing in vaccine design.

Reviewer 2 (Remarks on code availability):

I didn't run the code myself (no experience in Julia) there are very few instructions on how to prepare data (modify the JI file) or interpret the results. Examples of results in the readme would be helpful.

As mentioned above, we have now revised the GitHub repository to include additional documentation with tutorials and a full example with data of our standard workflow. The README now has a blue link directly to the web-based documentation.

Reviewer 3

Huhn et al propose a way to model bivalent binding of an antibody to surface bound antigens. Their model is based on the size of the ag as well as on parameters related to paratope-epitope interactions of the single arms. They show that their models allows for better modeling of apparent affinity and correlates well with experimental results. Moreover, they show that this model also correlates well with functional results of viral neutralization.

This study offer an interesting step beyond the current established literature, and provides an interesting and useful insight on the relationships between monovalent kinetics and functional neutralization. The conclusion is supported by both the model and the experimental results for two antigens. The ms is well written and easy to follow.

We thank the reviewer for taking the time to read our manuscript and for providing feedback.

Comments: The author conclude that the reach is determined by "both the antigen and the antibody". The antigen - they show that it is specifically determine by the size of the antigen. As for the antibody - the focus of the interface with the epitope and its kinetics. I wonder, however, what role, if any antibody specific parameters such as the flexibility of the hinges, the elbow and the H-L interface may play. The mechanism that the authors expose is structural and spatial in its nature. One may expect that if an antibody is somehow more flexibly and can separate its own arms more widely, the reach will also grow. This will have ramifications for antibody engineering as well. It will be interesting to hear the authors thoughts on this aspect of the reach.

We agree that the structure of the antibody contributes to reach and that it should be possible to modify antibodies or use synthetic bivalent molecules to manipulate the reach. A previous study (Shaw et al (2019) Nature Nanotechnology; <https://www.nature.com/articles/s41565-018-0336-3>) used a DNA origami platform to study the maximum antigen separation of IgM and all IgG sub-classes finding that they all tolerated a maximum separation distance of ~16 nm. Moreover, another study reached the same conclusion for IgE (Schneider et al (2023) ACS Nano; <https://doi.org/10.1021/acsnano.2c12647>). We therefore decided to examine the molecular reach of a different isotype, namely monomeric bivalent IgA antibodies (i.e. without the J-chain). We now find that the FD-11A antibody on the IgA rather than IgG1 hinge reproduces all of the same parameters, including molecular reach (Fig 2 above). Thus, antibody hinge does not appear to have a dramatic impact on binding properties. We have included text in the results and discussion along with a supplementary figure to explain this new result.

It would be interesting if the author discussed whether their observations about the correlation between the reach and neutralization potency can be generalized to other antigens (viral and others) and other types of antibodies (non-IgG1, single domain, etc) The study's conclusions may be depend on the structure/topology of the antigen. Since it was only tested on SARS-CoV-2 RBD Abs and one CD19 Ab- the limitations of generalizing the conclusions should be further emphasized.

We agree that the molecular reach, and bivalent binding more generally, may not be important for all antibody functions or for targeting all pathogens. We have revised the discussion to explain scenarios where molecular reach may not be important.

technical comments: In the steered MD when they pull antibody arms apart, could they estimate how much energy the steering imparts into the system? Is the estimated energy plausible under physiological condi-

tions?

We note that in the MD simulations we fixed one RBD biotinylation site and pulled on the second RBD biotinylation site (i.e. no pulling on the antibody itself). Importantly, this pulling is not meant to represent any physiological condition. Rather, the purpose of these simulations is to demonstrate that it is spatially possible for the antigen-antibody-antigen system to remain bivalently bound when the distance between the two biotinylation points is on the order of 30-40 nm.

We further note that the energy scale of native contacts within the coarse-grain model is approximate; the rigorous procedure employing replica-exchange simulations to carefully match experimental free energies is typically only possible when the system is <200 amino acids, with larger systems becoming computational intractable (see, for example, the rigorous procedure used in PMID: 30265800 to match an experimental free energy and an approximate method in PMID: 35654797). In the case at hand, even if we limit ourselves to one copy each of the antibody Fv (roughly 115 amino acids) and RBD (globular portion 196 amino acids), we significantly exceed this practical limit. All of this is to say that we cannot, in this instance, give a good approximation of the free energy barrier that must be overcome to unbind the antibody.

However, we can compute the pulling work during the steered molecular dynamics simulations (Fig. 3), which is generally on the same scale (several hundred kcal/mol) as the pulling work computed during all-atom steered molecular dynamics simulations with antibodies pulled off of SARS-CoV-2 RBD. Please compare, for example, Figure 3 B1 in PMID: 34228472 to Figure R1. We believe this result indicates that our simulations use a reasonable energy scale. However, due to the approximate nature of the native contact terms in the coarse-grain models used here and that this is not our key objective with this simulations, we prefer not to report the mean pulling work results in this work.

Figure 3: Mean pulling work computed over all trajectories that remain well folded for each of the six simulated antibodies for the native contact scaling factor that gives the maximum reach. Error bars represent the standard error about the mean.

How did they calculate Ab distance from the reference epitope? Are experimental ag-ab complex structures available for all tested antibodies or did they use models?

In the case of FD-11A, FD-5D, REGN10987, CD3022, EY-6A, and FI-3A structures are available (see Table S1) whereas for all other antibodies a competition assay combined with computational modelling was performed to estimate epitope location (see coloured circles on the RBD structure in Fig 5H and Fig S10A). To estimate epitope distance, we oriented the structure as shown in Fig 5H/S10, extracted the z-coordinate of each antibody, and subtracted it from the value of antibody 253 (reference antibody, see Fig 5H showing labelled reference antibody epitope location). We have added a sentence to the results to clarify how epitope location was identified for these antibodies.

Minor comments:

Figure 1C- graphical legend appears on 1E. It may be helpful to have a table with the average and stdev values for figure (1C) and related ones in the supplementary material.

The source data now includes an excel spreadsheet with all antibodies tested in the study along with their fitted binding parameters, neutralisation potency, predicted binding potency, etc. Together, this data can be used to reproduce all graphs in the manuscript.

Figure S1.C the plots at different Ag concentrations seem almost identical. Consider putting them on one plot to allow comparison.

The aim of Fig S1C is to show that they are nearly identical (i.e. that changing the antigen density and reach at the same time leads to the same level of antibody binding provided the formula in Fig S1B is used). We did initially plot them on a single plot but then its not possible to see that there are multiple conditions within the panel. Instead, we've subtracted them and placed plots of these residuals just below to show that they are nearly zero.

The Molecular dynamics reproduced impressively the calculated 'reach' values of antibodies that were co-crystallized with the antigens. It was important for this paper to support the discrepancy between the high calculated 'reach' values of this paper in comparison with the previous researches. The authors may want to provide representative MD files with the publication. All the files should be kept so that they may be provided upon request.

We have uploaded a sample of the MD simulations to FigShare (2.3 GB), which can be found at this link and now cited in our manuscript: Nissley, Daniel (2024). Coarse-grain MD input files and trajectories, antibody reach. figshare. Dataset. <https://doi.org/10.6084/m9.figshare.26828050.v1>

The measured correlation of the calculated 'reach' with the neutralization IC50 was good ($R^2=0.52$; $n=45$), and with all parameters (multiple linear regression) it was significantly improved ($R^2=0.82$; $n=45$). It is not clear if the linear regression was examined using cross validation. If not – I'd advise repeating it with CV.

As our aim was to simply compare the R^2 of each fit, we did not use the fitted model to make predictions. Therefore, we did not explore cross validation for any of the correlations produced in Fig 5. Importantly, the same potency data is used for all the datasets in Fig 5 and therefore, the R^2 can be directly compared across the datasets.

REVIEWERS' COMMENTS

Reviewer #1 (Remarks to the Author):

The authors addressed all comments satisfactorily. Thank you for a very interesting study.

Reviewer #2 (Remarks to the Author):

The authors have well answered all my points (and to the other reviewers). The manuscript is mature for publication, to my opinion.

Reviewer #2 (Remarks on code availability):

The authors have significantly improved the documentation on how to use the code with examples.